# Andean agriculture and hand tools: A qualitative approach of exploration of needs, barriers, and opportunities for innovation

**Liliana Cruz-Ausejo**[1‡*], **José del Carmen Abad Castillo**[2☯], **Claudia Cardenal**[3☯], **MD Zahid Hasan**[4☯], **Amit Bhattacharya**[4☯], **Jerome T. Galea**[5‡]

**1** Programa de Doctorado en Ciencias de la Salud, Facultad de Medicina, Universidad Nacional Mayor de San Marcos, Lima, Lima, Peru, **2** Escuela Académico Profesional de Tecnología Médica, Facultad de Medicina, Universidad Nacional Mayor de San Marcos, Lima, Lima, Peru, **3** Departamento académico de Arte y diseño, Pontificia Universidad Católica del Peru, Lima, Peru, **4** Department of Environmental and Public Health Science, University of Cincinnati, Cincinnati, Ohio, United States of America, **5** College of Behavioral and Community Sciences, School of Social Work, University of South Florida, Florida, United States of America

‡ LCA is the lead author and JTG is the Senior author on this work.
☯ These authors contributed equally to this work.
* ruth.cruz@unmsm.edu.pe

## Abstract

Peruvian Andean agriculture primarily consists of subsistence activities that rely on traditional methods. Rural farmers use locally made hand tools, often designed without technical or ergonomic considerations, increasing the risk of musculoskeletal disorders. This study examined farmers' views, needs, barriers, and support in hand tool design, and how they assess satisfaction with tools during potato harvesting. A qualitative study using a phenomenological design was conducted using semi-structured interviews with 13 rural farmers from Yanaca, Apurímac, Peru. A total of 25 interviews were conducted, audio-recorded, transcribed verbatim, and analyzed via a mixed coding strategy. Data was organized using a thematic approach and triangulated to validate coding. The results from the first guide were organized into five themes: task, human, product, qualitative and environmental. Some aspects such as social and techniques aspects for adopting hand tools emerged as well as enablers like economic accessibility, similarity to traditional tools. Six themes emerged from second guide covering tools conditions for efficient use, pre-work preparation, evaluation of the tool, consequences of a deficient tool, accumulated experience and appropriation; and family and gender dynamics. We found that hand tools used in high-Andean agriculture are shaped by technical features such as weight and handle size, and external aspects like climate effect and cultural familiarity, while material limitations, rough handle surface or inadequate tool size can act as barriers. In addition, efficiency of tools during work, preparation like a sharpened tip, fit between components head and handle, local maintenance facilitates farmers' satisfaction. These findings suggest that agricultural hand tool design should go beyond technical aspects to incorporate cultural and practical perspectives.

**Data availability statement:** De-anonymized data can be found in OSF repository: https://osf.io/fvr9t.

**Funding:** This work was funded by the Consejo Nacional de Ciencia, Tecnología e Innovación Tecnológica (CONCYTEC) and the Programa Nacional de Investigación Científica y Estudios Avanzados (PROCIENCIA) within the framework E077-2023-01-BM "Becas en Programas de Doctorado en Alianzas Interinstitucionales contest, grant number (PE501090201-2024-PROCIENCIA-BM) and the E033-2023-01-BM "Alianzas Interinstitucionales para Programas de Doctorado", grant number (PE501084306-2023-PROCIENCIA-BM). The sponsor had no role in the study design, data collection and analysis, decision to publish, or preparation of the manuscript.

**Competing interests:** The authors have declared that no competing interests exist.

## Introduction

Agriculture in Peru is highly diverse, shaped by complex geography, ecological variability, sociohistorical background, forms of productive organization, and uneven technological development [1]. Within this context, diversified smallholder family farming (DSFF) represents 95.7% of productive units. These units cultivate less than 2 hectares, often without certified seeds or modern technologies [2]. Predominantly located in high Andean regions, these smallholder units allocate around 66% of their temporary crops —such as potatoes and wheat—for self-consumption, serving as a primary food source [3]. Despite their key role in food security and the rural economy, DSFF farmers continue to rely primarily on manual agricultural tools [3,4] which may expose them to considerable health risks, particularly musculoskeletal disorders (MSDs) [4,5]. Although these manual tools are widely valued for their suitability to local farming conditions, evidence indicates that ergonomic risk is not determined by the absence of mechanization, but by the design of work systems [6].

Ergonomic risks may arise from repetitive movements and force postures associated with the design of tools, equipment and machinery [6]. Evidence from hand tool use in non-agricultural sectors show that manual work may involve significant risk when tool design and task characteristics do not align with human capabilities [7]. Commonly used tools by peruvian rural farmers included the *chaquitaclla* (Andean foot plough) [8], *allachu* (hoe), and various handmade picks. However, from a technical ergonomic perspective, these tools are not always adapted to farmers' physical characteristics, potentially perpetuating occupational hazards and limiting productivity [9]. Redesigning these tools through a contextual and participatory approach could mitigate these problems. Previous studies showed that modifying traditional agricultural tool based on anthropometric and ergonomic principles, such as adjusting tool length, weight, and handle design, can reduce physiological workload and improve task performance during agricultural activities [10,11].

Potato cultivation plays a central role in Peru´s food security system, as it represents the country´s main annual crop. In 2022, potato production accounted for 41.3% of all annual crops nationwide and is cultivated extensively throughout the Andean highlands, as well as in central and south coastal regions [2]. Given its national relevance and its strong dependence on manual labor during harvesting, potato production constitutes an important context for examining farmers' interactions with hand tools and ergonomic challenges, including sustained trunk flexion and repetitive movements.

Understanding user-centered design in the context of agrorural labor is essential for developing appropriate technologies that respond to the real challenges faced in the work field. However, there is limited research exploring the intersection between user perceptions, traditional tool usage, and the adoption of new manual technologies in rural agricultural settings. Innovating further on technological solutions must begin with a deep understanding of farmers' needs, perceptions, and practices. Farmers are not only end-users but also co-creators of innovation. Overlooking their knowledge, culture, and work practices undermines the adoption of new technologies even when they are technically functional [12,13]. To address this need for

co-creation, the "double diamond" design model frames innovation as two consecutive cycles of divergence and convergence. The first diamond covers Discover – Define (understanding, identifying real needs from the user's perspective and framing the problem), and the second diamond Develop-Deliver (developing and refining solutions) [14,15]. For example, Li applied this model to the service design of online agricultural product platforms, using the Discover and Define stages to identify user pain points and synthesize the core problem before proposing solutions directions [16].

In this study we focus on the first diamond, prioritizing identification and comprehension of farmers' needs and opportunities for tools improvement. This study address this gap through two linked questions:

1) What are the perceptions and needs of rural farmers that should be considered in the design of a manual tool, as well as the barriers and enablers to its adoption?

2) How do rural farmers perceive the use of traditional manual tools during potato harvesting in terms of satisfaction?

## Materials and methods

### Study design and methods

We conducted a qualitative study following an interpretivist paradigm and a phenomenological design to deeply analyze and interpret the meanings constructed by participants, reflecting their subjective realities [17]. Two semi-structured guides were used.

Interview guides: objectives and alignment with research questions:

Both guides explored hand tool use, they differed in analytical focus, the first emphasized contextual needs and design-related perceptions, whereas the second focused on experiential use and post-harvest satisfaction.

Guide 1 addressed research question one (exploring needs and perceptions to consider in manual tool design, as well as barriers/enablers to adoption). It covered: (i) Work context and harvesting logistics (Q1-Q3); (ii) Current tools and use practices (Q4,Q7); (iii) Perceived tool attributes and experience during digging, including comfort and perceived physical impacts (Q5,Q7,Q8); (iv) Contextual/environmental constraints shaping tools use (Q2, Q3,Q6), (v) Desired design requirements and adoption-related consideration, including reasons for accepting or rejecting a tool (Q9-Q17); and (vi) Sociocultural meaning of farming (Q18-19, see S1 File). In this study, perceptions refer to how individuals observe, understand, interpret, and evaluate an object, action, personal experience, or outcome [18].

Guide 2 addressed research question two (farmers perceived satisfaction with traditional hand tools during potato harvesting). It was administered after the harvest and focused on perceived in-use performance and any need for adjustment or repair (Q1), handling and comfort during use (Q2), perceived ergonomics of weight/size/shape and effort/strain (Q3), post-use condition and signs of wear (Q4), anticipated maintenance needs and perceived ease of maintenance (Q5) and overall satisfaction/ acceptability of the tool (Q6; S1 File)

The first and second interview guides were piloted with a local farmer, which allowed for adjustments in the sequence and wording of questions to ensure clear communication and cultural relevance. To facilitate expression and idea generation, we also used illustrations depicting the available manual tools, allowing farmers to visually indicate desired modifications. Additionally, paper and pencils were provided so participants could sketch and annotate their own proposals for improvement.

### Interviews setting and data collection methods

Data collection took place between Tuesday, May 13 and Saturday, May 24, 2025, in Yanaca district, Aymaraes province, Apurímac region, Peru, located at 3,315 meters above sea level and was conducted by a psychologist with expertise in qualitative methods. A prior meeting was held with the president of the local farming community, who authorized the study.

A purposive sampling strategy was applied to recruit 13 rural farmers. Participants were invited during the potato harvest based on referrals from key community contacts. Inclusion criteria included being 30 years or older and having at least six months of continuous engagement in agricultural work. A total of 25 semi-structured interviews were planned: 13 with the first guide and 12 with the second (one of the farmers could not be located after harvest). The selection of this number was based on the study by Hennink et al., which indicates that the range to achieved saturation in qualitative studies falls between 9 and 17 participants [19]. Interviews were held in familiar spaces, usually the participants' homes, starting around 6:30 p.m. after their workday. The first interviews lasted 30–40 minutes; the second, 15–20 minutes. A local Quechua-speaking interpreter participated, providing translation and cultural mediation in one interview to ensure comprehension and respect for the original language.

## Ethical considerations and informed consent

This study was approved by the Institutional Review Board of the Faculty of Medicine at Universidad Nacional Mayor de San Marcos, Perú (study code: 0072–2025) and by the Regional Health Directorate of Apurimac (DIRESA- Apurímac, in Spanish) under Letter N°004–2025-CRPDI-DEGDRH-DIRESA-AP, Registration N°1687. In addition, community-level permission was obtained from the community's formally recognized leadership. All participants provided informed consent prior to participation and verbally agreed to audio recording. There were no deviations from the established protocol.

## Data analysis

To ensure credibility, various methodological strategies were employed. Audio recordings were transcribed and independently reviewed by two researchers (LCA and EMC). A content analysis approach was used, involving systematic procedures to identify, code, and organize meaningful units within participants' narratives to generate contextualized and relevant interpretations [20].

A mixed coding strategy was applied. Initially, *a priori* set of analytic domains, informed by a thematic review of the human–hand tool system proposed for the design of manual tools, encompassing four key dimensions: human, product, task and qualitative variables [21]. These four domains were used as an organizing framework based on works by Jain *et.al* (Jain et al., 2018) to guide data familiarization and initial coding; however, specific codes were developed inductively from participants' narratives. As coding progressed, a coherent set of environmental conditions emerged as influential for tool use and perceived needs. We therefore incorporated an additional domain into the final analytic framework and results presented. The analytical framework presented in Fig 1 was informed by Interview Guide 1, which focused on perceptions, needs, and design-related considerations. Interview Guide 2 did not contribute to the definition of analytical domains; rather, it was analyzed separately to capture post-harvest experiences and satisfaction with traditional tools.

To enhance analytical rigor, each researcher conducted an independent analysis, and the findings were subsequently triangulated by investigators to ensure consistency and validity [22]. Pain-related codes were sub-coded by body region. We quantified each code as the number and percentage of participants mentioning pain in each region at least once (presence/ absence per interview), using a binarized code-document table. In addition, we quantified the hand tool characteristics and components which farmers had issues. Atlas.Ti v.25 software was used for data management and coding. Data saturation was achieved when no new codes emerged during analysis (in this case, after the 11th interview). The study followed the COREQ reporting guidelines for qualitative studies in support of rigor and transparency [23] (see S2 File). The anonymized data can be found in OSF repository: https://osf.io/fvr9t

To foster reflexivity, researchers (LCA and EMC) acknowledged their own beliefs and preconceptions throughout the analytic process. This allowed them to identify possible biases and enhance the transparency of findings.

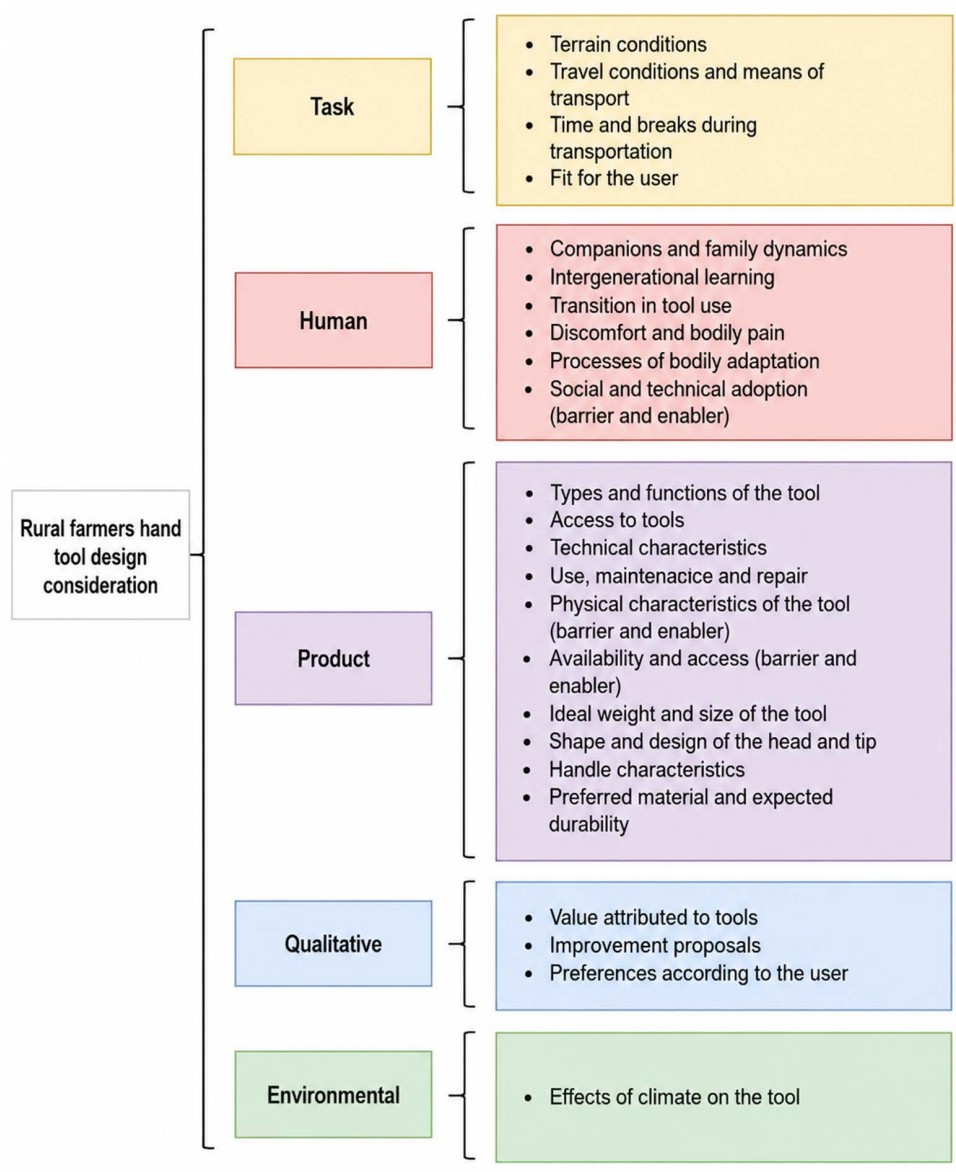

**Fig 1. Domains and codes for farmers' hand tool design considerations.**

### Inclusivity in global research

Additional information regarding the ethical, cultural and scientific considerations specific to inclusivity in global research is included in S3 File.

### Results

During the harvesting period in Llullasa, located 4.8 km from the district center, we conducted interviews with 13 persons (6 male, 7 female), aged 41–78 years (average height 155±8.8 cm). Data are presented according to the interview guide used, and illustrative quotes for each of the domains (additional quotes are presented in S4 File). The first interview guide

addressed participants' perceptions and needs that should be considered in designing a manual tool, as well as the barriers and facilitators for its adoption. Domains and codes are presented in Fig 1. Names have been changed to conceal the identity of participants.

### Task category

**Terrain conditions.** Potato harvesting occurred between May and June, under cold and dry conditions, (10°C to 20°C), which was favorable for the activity. Farmers wore light clothing, such as long-sleeved shirts, trousers, and hats or caps. However, the workers reported difficulties accessing the fields located on slopes reachable only on foot. Another relevant aspect reported by farmers was the condition of the land, as it determined how physically demanding the work was, influencing the ease of using tools (Fig 2).

*"The soil in that area is clayey […] and somewhat red… when there's a lot of moisture in the soil, it's soft, and when it starts to dry, the soil becomes hard" (p9, man)*

**Travel conditions and means of transport.** Traveling to the cultivation fields began in the early morning, between 6:00 a.m. and 8:00 a.m. Tools, food, and sometimes young helpers were transported on pack animals: horses, donkeys or mules. Those who did not have access to support must carried all their belongings themselves (Fig 3)

*"I tie it to the horse…there are some ropes to hold it […] if you do not have that, you must put the old clothes you have, arranging them on the horse's back. The horse carries the pots where you can cook potatoes […], I carry my baby there on the horse" (p11, woman)*

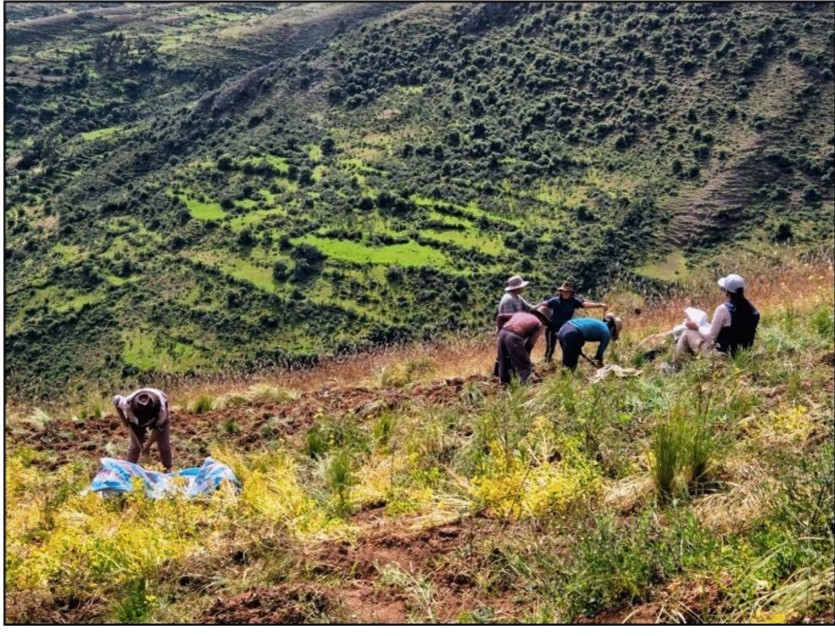

**Fig 2. Farmers during potato harvesting (photo taken by the author).**

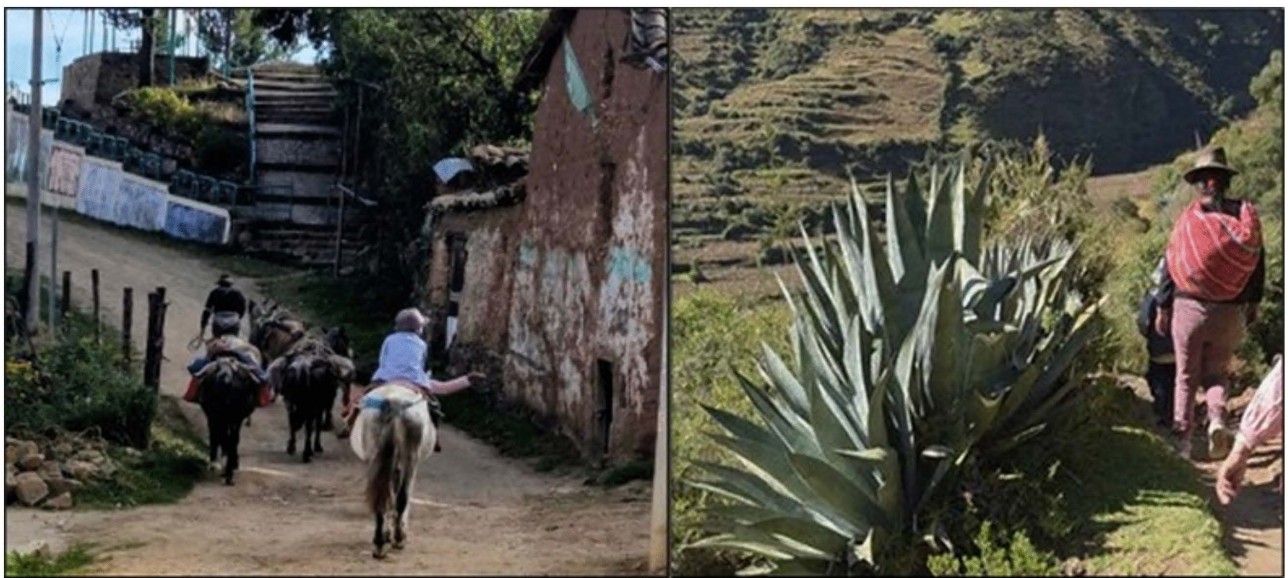

**Fig 3. Transport of tools: using pack animals (left), on foot (right).** (photos taken by the author).

The journey, which took between 1–2 h, included breaks, reflecting the physical exhaustion associated with the distance and time required to reach their fields. Although these breaks provided temporary relief, some workers avoided them due to the urgency of completing their tasks

*"It takes me an hour and a half…by the long path with the horses […] I must go quickly to keep up with harvest, otherwise what time would I finish? I don't get to rest" (p7, woman).*

**Fit for the user.** During the potato harvest, farmers adapted their tools primarily through physical modification of existing hand tools, prioritizing size, weight, and fit. Tools that could be resized, enlarged or repaired, were especially valued, as they were perceived as personal and tailored.

*"…because it's a bit lighter, and, uh… not heavy […] it fits your hand, a bit smooth, not rough […] you already know your tool, right? Your hand itself already feels that it's made for you." (p6, man)*

## Human category

**Companions and family dynamic.** The harvest is a community and family activity involving men, women, and children. Children used small tools to gradually join fieldwork, and families alternated work shifts to provide mutual support during the harvesting process.

*"We call it Ayni; in the community everyone works together. There are no individual family plots; all the work is collective, is a communal workday"* (p9, man)

Although the participant described agricultural work as fully collective under the Ayni system, land tenure and labor organization in the Andean Highlands were more heterogeneous. Most residents cultivate owned plots that were socially

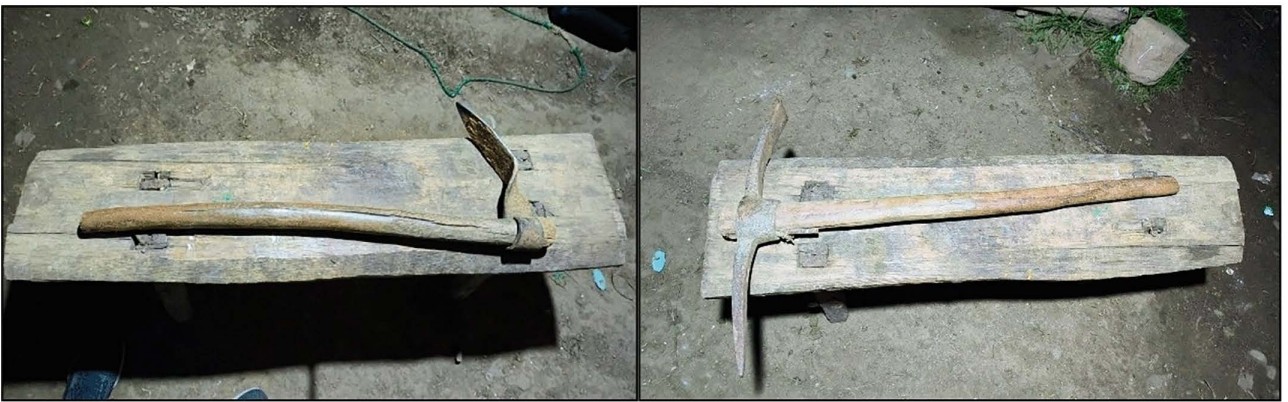

recognized by the community and regulated through titles. However, a proportion of households lacked formal landownership and therefore shared and rotated access to communal parcels. This diversity was also reflected at the national level: in 2022, only 35.5% of agricultural producers held a formal property title for at least one parcel, while 64.5% lacked land titles, shaping varied land-use and labor arrangements [2]

**Intergenerational learning.** The transmission of knowledge regarding the use of agricultural tools occurred primarily through intergenerational means, via observation and imitation between parents and children, allowing them to progressively engage in this task.

*"You always see your dad working, and you also want to be there, helping like that […] watching your dad working in the field"* (p12, man)

**Transition in tool use.** Regarding the tools, although the *allachu* (hoe; Fig 4) was used by elder generations, it has been replaced by the pick nowadays, persisting only in certain cases due to family tradition or as a cultural inheritance.

*"I still remember my dad when I was a kid. Back then, there were no picks; my dad only used the allachu. Then, little by little, picks appeared […] and now there are no allachu left."* (p9, man)

**Discomfort and bodily pain.** Farmers described experiencing discomfort and pain in the arms, wrists and lower back while harvesting and using hand tools, highlighting the physically demanding nature of their activity.

*"When you work constantly or spend all day on something heavy, the next day everything hurts"* (p11, woman)

Table 1 summarizes the distribution of self-reported bodily pain by body region during hand-tool use. The most common reported pain region was the lower back (9/13; 69.2%), followed by arms (6/13; 46.2%) and shoulders (4/13; 30.8%). Pain in the elbows and hands were reported by three of 13 participants, whereas neck and fingers pain were less frequent. When stratified by sex, women reported elbow and hand pain more frequently compared with men; lower-back pain remained high in both groups.

**Processes of bodily adaptation.** Farmers perceived discomfort and pain as a natural aspect of their work. Over time they reported adapting by incorporating these sensations into their daily routines and managing them through brief rest periods. Participants reported that, in this community, potato harvesting during the study period typically lasted around seven

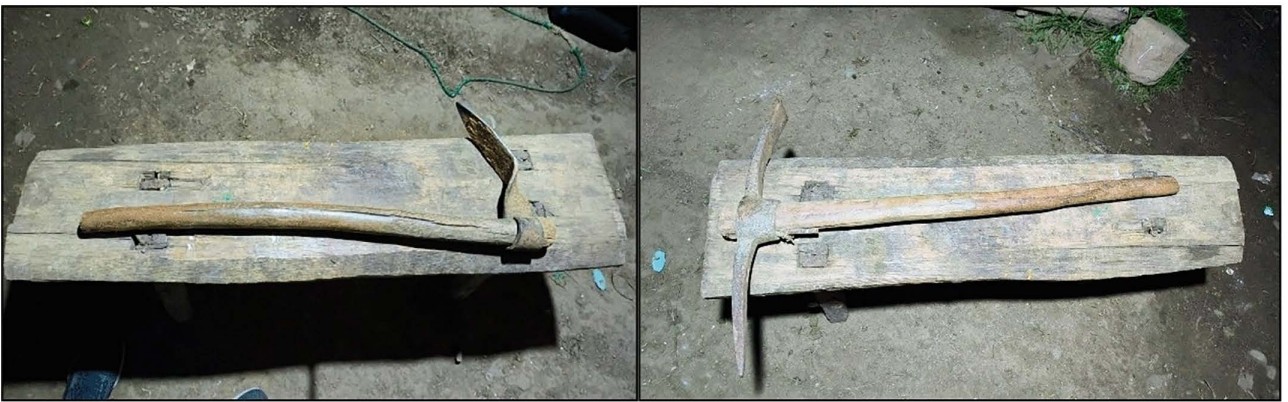

**Fig 4. Traditional hand tools: *allachu* (left), pick (right).**

**Table 1. Self-reported musculoskeletal pain by body region during hand tool use overall and by sex.**

| Body region | Total n=13 | Sex | |
|---|---|---|---|
| | | Men n=6 | Women n=7 |
| Neck | 1 (7.7) | 1 (16.7) | 0 (0) |
| Shoulder | 4 (30.8) | 2 (33.3) | 2 (28.6) |
| Arms | (46.2) | 3 (50.0) | 3 (42.9) |
| Elbow | 3 (23.1) | 0 (0) | 3 (42.9) |
| Hand | 3 (23.1) | 0 (0) | 3 (42.9) |
| fingers | 1 (7.7) | 0 (0) | 1 (14.3) |
| Lower back | 9 (69.2) | 5 (83.3) | 4 (57.1) |
| Knees | 3 (23.1) | 2 (33.3) | 1 (14.3) |
| Non-specified | 2 (15.4) | 0 (0) | 2 (28.6) |

n: number of participants who mentioned pain in that body region at least once.

Participants could report more than one region.

consecutive days, with workdays extending from approximately 8:00 a.m. to 5:00 p.m. This sustained and repetitive exposure was described as physically demanding and contributed to cumulative fatigue and bodily strain during harvest period.

*"Sometimes you're not used to this… you feel a bit tired; your spine is affected"* (p6, man)

*"The first day your spine hurts, the arms…you have to get used to"* (p11, woman)

**Social and technical adoption (barrier and enabler).** One aspect highlighted was the influence of social and technical factors on the adoption and use of hand tools such as familiarity, local maintenance, the use of local materials and the resemblance to traditional tools.

*"…that tools were made from local woods, from Tasta or hardwoods or lloje"* (p3, man)

*"For usefulness… it should always resemble the pick"* -referring to the pick head (p11, woman)

## Product

**Types and functions of the tool.** Farmers considered two manual tools for potato harvesting: the allachu (hoe), a traditional single-pointed digging tool and the pick (Fig 4), with a pointed end and a blade for loosening soil, extracting potatoes and removing stones or clods. The pick was appreciated for its versatile features in terrain: digging, extraction and removing soil.

*"The pick has two points, but the allachu has only one. […] The pick has two functions: this flat side (referring to the blade) can open the soil a bit wider […] and this pointed side is for digging and pulling out stones"* (p6, man).

**Access to tools.** Both hand tools featured locally crafted wooden handles (*tasta* or *chachas* wood) and metal heads, adapted to the terrain and considered as more practical than mechanization on slopes. The *allachu* was used by elders and traditionally crafted with wooden handles shaped by farmers and metal heads forged by the blacksmiths. When tools were not available, they were either shared among community members or improvised with materials at hand. After work, tools were stored at home or buried near the fields for safekeeping.

*"Of course, we make them ourselves for our own use, for what we need.", "Nowadays they don't make allachu anymore, just like that; I mostly find picks, I don't find allachu anymore"* (p6, man)

**Technical characteristics.** In harvesting, lightweight and slender tools were required, with metal heads featuring thin edges to prevent cutting the tubers. Deep digging techniques were considered essential, as shallow excavation increased the risk of damage. In the case of picks, the use of the point is valued over the blade, while curved handles were considered impractical for the task.

*"The pick should be medium-sized, so you don't get tired, right? Sometimes it's big or small, and if it's smaller you get tired from bending over"* (p10, woman)

*"With the tip of the pick, it's so you don't split it; since it has a tip, you don't break the potato."* (p13, man)

**Use, maintenance and repair.** The tools used show natural wear resulting from climatic and working conditions. Their restoration and adjustment were carried out locally, often with the assistance of blacksmiths, to extend their lifespan and preserve their functionality.

*"Sometimes, after working extensively on the pick blade, it loses its sharpness. It needs to be sharp, so we sharpen it ourselves; if it becomes completely worn out, then we take it to the blacksmith to be repaired, so we can continue working. This is something we always do."* (p7, woman)

**Physical characteristics of the tool (barriers and enablers).** Participants identified physical characteristics of the tool that functioned as barriers or enablers for their adoption and continued use. Among these, weight (average of 3.0±0.2 kg; mean±SD, n = 12), emerged as key attribute influencing whether farmers retained, replaced or sought alternative tools. Tools perceived as excessively heavy discouraged continued use, whereas overly light tools were considered functionally insufficient, leading participants to prefer an intermediate weight that enabled effective soil penetration. Other relevant characteristics included handle surface and shape.

*"There are other rough handles that make your hand hurt and are uncomfortable […] they have splinters that damage your hand, of course […] they give you blisters."* (p6, man)

*"…There's a small pick, but it doesn't dig in well. I usually use this large pick, but it's too heavy […] I'm thinking of buying one like this."* (p5, woman)

Table 2 presents the measured physical characteristics of the hand tools: picks (n = 10) and allachu (n = 2). Overall, the pick showed higher values for weight, total length and handle length compared with the allachu. The average handle diameter of both tools was 39.75±3.1 mm, however across tools it differed by approximately 3 mm (40.3 vs 37.3) mm. Head dimensions also differed: the pick had a tip length of 222±32.2 mm and a blade length of 224.5±35.9 mm, whereas the allachu, having a single tip, showed a tip length of 200 mm and a smaller head junction. Based on the anthropometric

**Table 2. Physical and ergonomics characteristics of hand tools.**

| Hand tool type | Weight (kg) | Total length (mm) | Handle length (mm) | Handle diameter (mm) | Tip length (mm) | Blade length (mm) | Head juntion width (mm) |
|---|---|---|---|---|---|---|---|
| Pick | 3.1±0.2 | 895.7±63.6 | 835±84.5 | 40.3±3.2 | 222±32.2 | 224.5±35.9 | 93.8±4.9 |
| Allachu | 2.7±0.3 | 822.5±81.3 | 720±84.9 | 37.3±0.4 | 200±0.0 | | 80.0±14.1 |

data of the sample (S5 File), the mean maximum grip diameter for women was 39.7±2.3 mm, whereas for men it was 43.3±3.1 mm. These findings indicated that the current tool handle diameter was suitable for 67% of the participants, corresponding to 83.3% of men and 50% of women.

**Availability and access (barrier and enabler).** The availability of tools, such as picks, through local hardware stores or local vendors was reported as facilitating their use within the community. Participants also mentioned that tool's affordable cost contributed to their prioritization and adoption.

*"If you have the allachu, well, you take the allachu. If you have the pick, you take the pick… but mostly it's the pick we buy at the hardware store". A pick at the hardware store costs around 25 to 35 soles"* (p11, woman)

**Product consideration: weight, size, handle, material characteristics and durability.** Farmers highlighted key features of hand tools, including their weight, size, and the appropriate shaping of the head and tip to enable comfortable use. Tools should not be excessively heavy, as this may hinder the work of women and older adults. At the same time, they must be made of durable materials, such as dry wood and metal, that could withstand impact

*"Because of the weight, it should be light so you can carry it far" (p13, man)*

*"Women use smaller picks and men use slightly bigger ones… the pick is light for men, but for other tasks, as I was saying, they're a bit larger" (p9, man)*

*"The tip should have a sharper point. The head should be longer." (p13, man)*

*"A handle that's smooth, you know, for your hand, so it's a bit soft and not rough", "Sometimes this handle is too short; I'd prefer the handle to be a bit longer, you know? So, it can reach further […] it needs a slightly bigger handle." (p6, man)*

Table 3 presents the main characteristics and components of the tools in which farmers reported inconveniences or conflicts. The most frequently cited issues included the curvature of the handle and inappropriate handle length (either too long or too short). Regarding tool weight, 57.1% of female farmers reported that the tool was excessively heavy for them. In contrast, 33.3% of male farmers indicated the need for repairs and surface improvements, as the presence of splinters required sanding to ensure safe use.

**Table 3. Tool characteristics, components and reported inconveniences by gender.**

| Tool characteristics and components | Total n=13 | Sex | |
|---|---|---|---|
| | | Men n=6 | Women n=7 |
| Tool perceived as excessively heavy | 5 (38.5) | 1 (16.7) | 4 (57.1) |
| Tool perceived as optimal weight | 5 (38.5) | 2 (33.3) | 3 (42.9) |
| Handle characteristics | | | |
| Needs repair | 2 (15.4) | 2 (33.3) | 0 (0) |
| Curved handle | 2 (15.4) | 2 (33.3) | 0 (0) |
| Handle too long | 1 (7.7) | 0 (0) | 1 (14.3) |
| Handle too wide | 1 (7.7) | 1 (16.7) | 0 (0) |
| Handle too short | 4 (30.8) | 2 (33.3) | 2 (28.6) |

n indicates the number of participants who mentioned the characteristic.

### Environmental

**Effects of climate on the tool.** In the Andes highlands, climatic factors such as humidity, rainfall, solar radiation, and soil type negatively affected the maintenance and preservation of tools, reducing their useful life. Among these, humidity and rainfall stand out as the conditions that affected their preservation. About the handle a farmer said: *"rain makes it rot!"*; and about the iron head, he added: *"ah, that rusts"* (p4, man)

### Qualitative

**Value attributed to tools.** Farmers described a sense of familiarity and attachment to their working tools developed through everyday use. The importance of a lightweight handle to facilitate use is highlighted: *"I like when the handle is smooth… not so heavy"* which was associated with ease of handling rather than performance alone. In addition, the *allachu* was recognized as an object of special significance and value with some participants noting that *"the allachu is kept as a relic",* even when it was no use regularly. This relationship extends beyond functionality, reflecting familiarity and continuity over time.

**Improvement proposal and preferences according to the user.** The aspects mentioned led participants to express their wishes and suggestions for improving the physical and technical characteristics of the tools. The proposals were recorded in sketches made by the participants (S6 File). Preferences leaned toward lightweight, sharpened and medium-sized instruments, with these considerations frequently appearing in the design improvement proposals.

*"A tool a little easier, that doesn't weigh too much because I'm older, and it would be helpful if it weighed less."* (p3, man).

*"This head should be sharper… or made bigger, so that with one strike, with two strikes, it would go deeper into the soil."* (p13, man)

*"Between the allachu and the pick…the two-pronged pick […] I would prefer it because with the pointed end you can dig, and with the blade you can pull"* (p7, woman)

The following section addressed the second research question, aimed at understanding the perceived satisfaction of farmers regarding the use of traditional hand tools. The resulting categories and codes are presented in Fig 5, and additional representative quotes are presented in S7 File.

**Tools conditions for efficient use.** This category covered the technical and material aspects affecting tools efficiency during potato harvesting. While weight was initially identified as a physical attribute influencing tool choice (product dimension), its relevance became more evident through farmers' in-use experiences. Farmers associated satisfaction with efficiency, which depended on balancing attributes such as weight: heavier tools increased fatigue and bodily strain, whereas lighter tools reduced effectiveness in soil loosening. Perceptions of optimal weight, varied according to individual characteristics. Additionally, keeping the tool head sharp and well-maintained was considered essential for sustained performance with minimal effort.

*"Let's see, one of them is a little heavy, I look for a small pick that isn't heavy, that's the one I have to take. Light"* (p7, woman)

*"Normal, right? It's not very heavy anyway", "Since it's very light, I haven't gotten tired"* (p3, man)

*"I'm going to take it to have it sharpened… to make the tip sharp again"* (p1, woman)

*"With so much work, the tip wears down, so I have to take it to the blacksmith to fix it, I mean, to sharpen the point"* (p3, man)

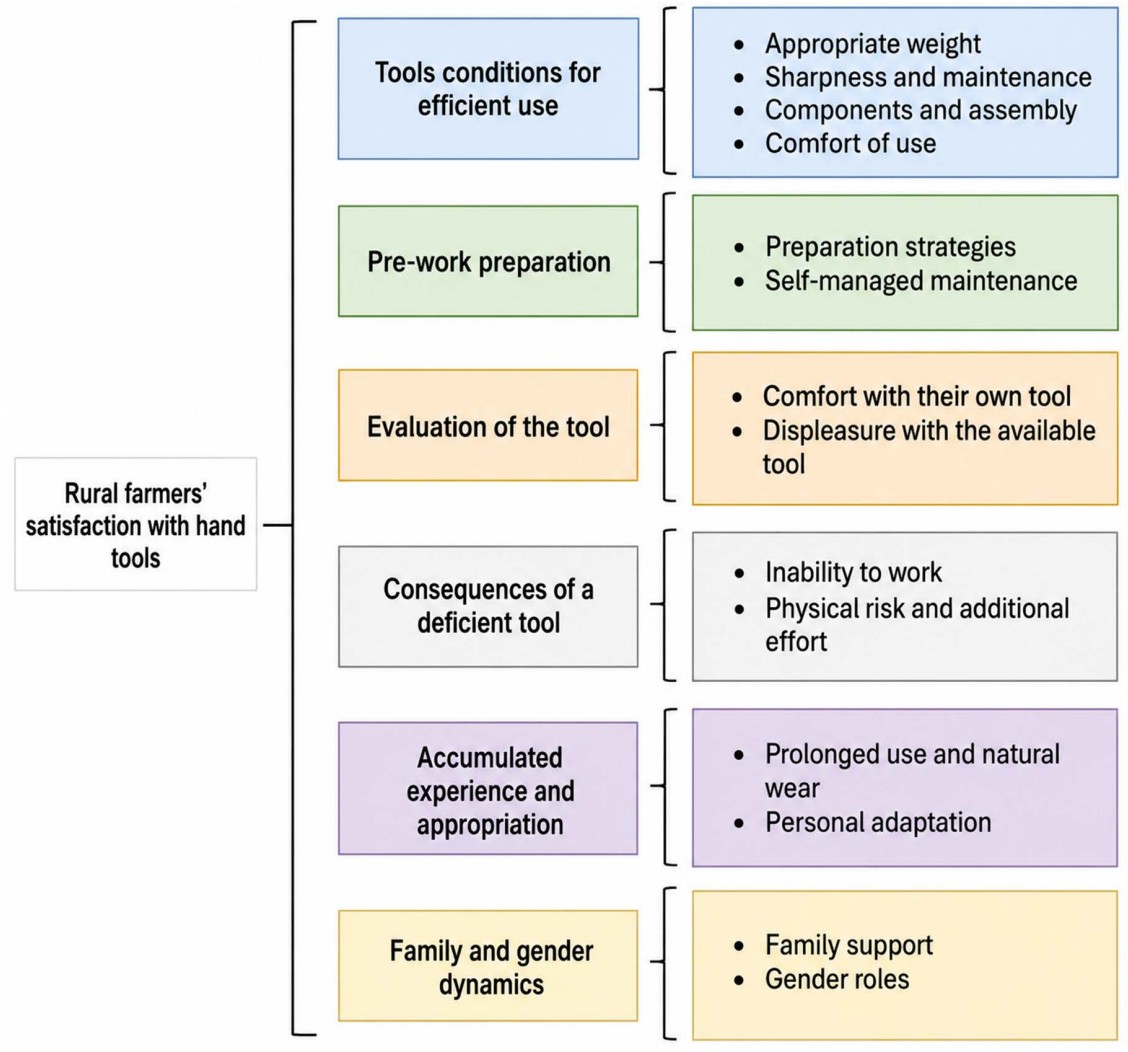

**Fig 5. Domains and codes for rural farmers' satisfaction with hand tools in potato harvest.**

Farmers emphasized not only the materials of their tools, such as the wood of the handle or the iron of the head, but also qualities like origin, strength, and durability. They considered the proper fit between components, especially handle and head, essential to avoid detachment and ensure control. Likewise, comfort was a key criterion, understood beyond ergonomics to include familiarity and adaptability for each worker.

*"You need to put more little pieces of wood in this spot to tighten it. Some woods, when they dry… If the head loosens, it's because of the wood" (p2, men)*

*"The handles sometimes loosen with use. They put in a wedge, little sticks or nails; others use rubber to make it fit properly" (p12, men)*

*"No, all good. I haven't felt any discomfort" (p1, woman)*

*"…something that bothered me was that it was kind of shaking (loose), since I work with it every day…" (p5, woman)*

**Pre-work preparation.** This category referred to preparatory practices that farmers performed to anticipate failures and optimize tool use, demonstrating their technical self-management. These actions varied by task but commonly included drying the handle before attaching it to the head to avoid looseness, and making temporary adjustments, such as wedges, to extend the tool´s life and ensure safety.

*"Of course, if we're going to the harvest, we must change the handle. How is this one, old? Already worn, right? After a year it must be replaced; it must be new"* (p7, woman)

*"My tool, I have to take it ready, yes, you have to make sure it doesn't come loose"* (p10, woman)

*"…yes, we have to bring wood. I look for it, let it dry, cut it… we also scrape it and attach it to the head"* (p4, man)

*"The metal head had come loose. It was something that made noise, so I was hitting it on the stone so it wouldn't move"* (p5, woman)

*"Just hitting it… maintenance isn't that difficult"* (p6, man)

**Evaluation of the tool.** Evaluation emerged in the field interviews as a core aspect of farmers' satisfaction, encompassing both functional performance and the comfort/displeasure using hand tools. Positive assessments arose when tools met user's expectations and working conditions. However, tools were not uniformly accepted, as specific design features appeared to influence the performance and ease of use.

*"Everything was fine, there were no problems"* (p2, woman)

*"It has been very smooth, and there were hardly any failures, it did a great job!"* (p13, man)

*"Yesterday I took aunt Sandra's pick, right? But it bothered me a little and I don't know, I didn't like it… I couldn't dig properly, so I just used my own pick. I don't know if was the tip of her pick that felt very strange… hers had a wide tip"* (p10, woman)

*"…there was a pick that Mr. Jose brought, it was wider. But that pick, since it's wide, can break the potato; it must be a thin pick"* (p12, man)

**Consequences of a deficient tool.** This category referred to the negative effects of poorly maintained tools, including fatigue, physical discomfort, and wear caused by loose handles, worn tips, or shifting heads during work. Beyond structural flaws, these issues exposed the limitations of daily labor and the physical risks of a poorly fitted head, generating concern among workers.

*"What happens if the metal head falls in the field? In the field, the handle breaks, and you can't work"* (p7, woman)

*"When it's blunt, I mean it's not sharp, it doesn't go into the soil"* (p5, woman)

*"It's already worn, and in these areas, since you're holding it as it's worn, your hand itself starts to hurt"*, *"If they don't sand this part (handle) well, little splinters remain, and that bothers your hand and sticks into your palm"* (p10, woman)

**Accumulated experience and appropriation.** This category encompassed farmers' narratives regarding the prolonged use of their tools and the relationship they develop with them over time. As they consistently use their tools, farmers developed increasing familiarity with their tools, which shaped their confidence and preference during use.

*"Yes, just now when I hold it, I also felt it move a little (hitting sounds), it's loose"* (p4, man)

*"Yeah, this is a little bit it loosens where the metal head is"* (p6, man)

*"For my liking, it has to be tight, otherwise I can't work"* (p7, woman)

*"The first days (…) when you get a pick made like this, while you're working you feel that it's a little heavy or not to your liking, and then you have to fix it"* (p10, woman)

**Family and gender dynamics.** Perceptions and practices related to the maintenance and repair of tools were distributed according to gender within the farmers' family and work contexts. The narratives indicated that men assume technical repair tasks, which ranged from obtaining handles and commissioning the manufacture of specific parts to anticipating potential failures.

*"My husband, here, almost everyone does their own maintenance; they put on the metal head and the little stick themselves"* (p10, woman)

*"The men usually do it, I don't know how, but they have to scrape the metal head"*, *"My husband takes care of these things, makes sure everything is fine…"*, *"When they have time, they sand it, sand it, measuring it, until it reaches the right size; once it's ready, they just leave it as is"* (p10, woman)

## Discussion

Farmers identified five key aspects in tool design: human, product, task, qualitative and environmental areas. These dimensions emerged from the participants' narratives and align with Jain´s et.al (2018) proposal for improving non-powered hand tools to prevent work-related problems [21], while adding environment as a new theme within their narratives (Fig 1).

Farmers placed a central emphasis on technical characteristics such as weight. In our measurements, tool weight was approximately 3 kg. General hand tool ergonomics guidance recommends keeping tool weight as low as practicable and cites 2.3 kg as an upper value to help reduce fatigue [24]. Tools perceived as too heavy were described as increasing fatigue in hands, arms and shoulders and limiting use by women and older adults, whereas an excessively light tool were perceived as ineffective for breaking soils. Although lightness was valued, participants emphasized the need for a functional balance. In clay soils with compacted clods, the complementary use of pointed and flat ends was perceived as more effective despite the additional weight. In addition, inadequate dimensions can reduce effective force transfer, requiring greater impact speed and thereby increasing the risk of slippage and injuries [25,9]. These findings suggest that tool weight should be carefully balanced in future design efforts, minimizing fatigue while maintaining sufficient mass to ensure effectiveness under demanding soil conditions.

Regarding the handle, its shape and finish are critical for grip comfort and injury prevention. A small or poorly finished handle may slip, whereas an appropriately design handle can provide greater safety [9,26]. Grant et.al. [27] noted that handle diameter directly affects grip strength and, consequently, the muscular effort required. In this study, handle diameter measured at the primary grip zone ranged from 37.3 to 40.3 mm with a general diameter of 39.75 ± 3.1 mm. Based on the sample anthropometric data, the handle diameter was compatible with 67% of the participants (83.3% of men and 50% of women), suggesting a closer fit for male users in this sample.

Tool length is also relevant, the Canadian Centre for Occupational Health and Safety (CCOHS), indicates that total tool length with the blade placed on the ground should be approximately at the user's elbow height [28]. While national anthropometric references values for Peru are limited, Escobar [29] reports estimated standing elbows height of approximately 103.1 cm for men and 95.0 cm for women. However, anthropometric variability in Andean populations should also be considered. Evidence from highland Peru shows relatively shorter limb segments under conditions of environmental

stress, reflecting growth trade-offs that prioritize essential functions over limb length [30]. This suggests that ergonomic guidelines derived from non-Andean populations may not be directly applicable and could result in tools that are suboptimal in length for local users. In our study, the measured total tool lengths were generally below the reference values, which may increase bending during harvesting tasks. When considered alongside the pain distribution reported by participants, particularly the presence of hand and finger discomfort among women, and the frequent reporting of lower-back pain, these measurements suggest that handle-hand fit and tool length are important aspects to consider in future redesign efforts. The testimonies collected in this study reinforce that an appropriate diameter, shape and tool length contribute to perceived control and familiarity during use and are closely linked to comfort and physical strain.

Another relevant feature is the design of the tool head and tip, which directly influences potato protection and efficiency when working in clay soils. In this regard, farmers expressed a preference for sharp tips and even multiple and interchangeable heads, emphasizing the need for a flexible tool design that responds to crop-specific practices and farmer´s operational needs. Durability also emerged as a central factor: while local woods such as *tasta* are valued for their strength, others like eucalyptus are perceived as fragile. Although most handles are made from local wood, both quality and design are crucial, as inadequate configurations increase the likelihood of injury [25,9]. Additionally, climatic conditions such as humidity and rainfall may accelerate handle deterioration, promote head detachment, and increase the frequency of local replacements. Tool degradation may also result from mechanical wear caused by repeated contact with soil and its conditions including the presence of rocks, in addition to corrosion processes such as rust formation. In agricultural contexts, these abrasive interactions can contribute to the wear of tool edges, which may explain the frequent need for sharpening reported by participants.

Likewise, barriers to the adoption of hand tools were identified, including poor finishes, dull tips, or short handles that increase fatigue. Economically, the need for constant maintenance, such as sharpening, represents an additional cost that limits accessibility. Social factors were also observed to hinder the adoption of innovations as tools that differ from traditional models are often perceived as unfamiliar, which can reduce acceptance [31]. Beyond these, a previous study has discussed a perceived tension between comfort and productivity in agricultural work. In this context, some farmers may associate heavier or more demanding tools with greater effectiveness, even if they increase physical strain. While this study did not directly assess productivity outcomes, this perception may help to explain resistance to adopting more ergonomically tools, despite their potential benefits for reducing fatigue over time [32].

In contrast, enablers emerged, such as the possibility of local repairs through community blacksmiths or the use of accessible replacement parts. A particularly relevant factor is the ergonomic fit of the handle, as adaptation to the hand not only enhances comfort and safety but also promotes adoption [33]. However, fieldwork revealed that blacksmithing is not a full-time occupation in the community, but rather a secondary activity performed by local farmers upon request. As a result, tool reparability depends on the limited availability of these individuals, which may delay repairs and reduce accessibility. Under these conditions, commercially available replacement parts and imported tools may contribute to transition toward externally manufactured tools. In this context, local organizations or agricultural extension services could play a role in supporting the preservation of existing blacksmithing practices through training and technical reinforcement, helping sustain locally adapted repair systems.

In relation to the second question, farmers' satisfaction emerged as a complex phenomenon, resulting from the balance between functionality and subjective appropriation. A central aspect is comfort of use, which encompasses adequate weight, a firm grip, and a secure assembly. These elements align with FAO recommendations, which identify one of the main problems of hand tools as the use of soft, low-quality wooden handles [34]. Another key factor is the possibility of performing self-maintenance, such as sharpening the tip, adjusting the handle, or replacing damaged parts, which reinforces a sense of control over the tool. These practices are grounded in farmer's experiential knowledge and may be further supported by basic technical criteria, to enhance their effectiveness [34].

Satisfaction was also linked to familiarity built through experience, often transmitted intergenerationally. This form of learning, which in many cases acts in place of formal training or written instructions, strengthens both appropriation and the bond with the tool. In this regard, dissatisfaction may not necessarily stem from tool design itself but rather from the absence of adequate training that would normally accompany the introduction of externally developed tools within a more formalized context [21]. Nevertheless, in the rural agricultural context, characterized by informality and the lack of systematic training structures, pragmatic learning ensures the continuity of the craft and reinforces a sense of belonging, while potentially perpetuating technical limitations that affect both efficiency and satisfaction. These findings highlight the importance of participatory design approaches, such as the double diamond model, that integrate local knowledge with technical design criteria to ensure that innovations are both contextually appropriate and functionally effective.

Although previous studies have emphasized the importance of weight and ergonomics in tool design [25,33], less attention has been paid to aspects such as local reparability, intergenerational learning, and the symbolic dimensions of tools [35]. These differences may be explained by methodological approaches and by the indicators prioritized by workers, particularly productivity [12]. In this study, these factors are further shaped by ongoing demographic and technological changes. Field observations indicate that the agricultural workforce is predominantly composed of middle-aged and older adults [36], as younger generations increasingly migrate to urban areas or mining-related employment. Within this transition, manufactured tools have gained importance, while older, locally crafted implements, such as the allachu, are increasingly rare. Though traditional knowledge persists, farmers have shifted toward tools such as the pickaxe due to their perceived versatility, particularly the dual-function head. At the same time, the widespread availability of imported tools and replacements parts through local hardware stores offers faster access compared to locally made or repaired tools, whose production depends on limited local craftsmanship.

These findings highlight that the adoption of innovations in agriculture does not depend solely on the introduction of improved techniques driven by technological feasibility [12,37], but rather on recognizing the complex interaction between technical characteristics, durability, accessibility, climate, and culture. In this context, tool sustainability is closely linked not only to design features but also to their capacity to be adapted and repaired within local systems. Approaches such as participatory ergonomics and co-design, which integrate workers' needs and local capacities are therefore essential to ensure that hand tools remain usable and valued over time [35,37].

## Strengths and limitations

These findings provide substantial insights into the needs, perceptions, barriers, and satisfaction of rural farmers regarding their use of manual tools. As part of a broader research project aimed at establishing an initial baseline from a cultural perspective, the results help identify opportunities that inform not only the design and adaptation of technologies to rural contexts but also reflect on the social and cultural ties that shape the relationship between communities and their tools.

Although this study examined tools that are ubiquitous in the Andes within potato harvesting, one of the most common smallholder farming activities in this context, the sample was restricted to a single agricultural community. This may limit the transferability of the findings to other crop systems, agricultural tasks, or Andean communities with different environmental conditions and tool repertoires not documented in this article. In addition, resource constraints prevented extending data collection to multiple communities, which would have strengthened broader applicability.

## Conclusion

This study highlighted that the use and adoption of hand tools in high- Andean agriculture are shaped by a combination of technical, ergonomic, environmental, and sociocultural factors. Farmers emphasized the importance of tool weight, handle design, length and material durability, in relation to comfort, efficiency, and physical strain during potato harvesting. Rather than prioritizing minimal weight alone, participants value a functional balance that allows effective soil loosening while limiting fatigue, with variations according to sex, age and physical capacity. Beyond physical characteristics, tool adoption

was influenced by familiarity, perceived efficiency, and the possibility of adjustment and maintenance. The ability to repair or modify tools locally, whether through self-maintenance or access to community-based repair skills, emerged as a key supporting continued used. In contexts where local repair capacity is limited, commercially available tools and replacement parts increasingly shape farmers' choices.

The implication of these findings suggest that agricultural tool design should move beyond technical considerations to incorporate local work practices, user diversity, and repairability with rural systems. Approaches such as participatory ergonomics and co-design, may contribute to more sustainable and context-appropriate hand tools in high-Andean agriculture.

## Supporting information

**S1 File. Semi-structured guides (1 and 2).**
(DOC)

**S2 File. COREQ guidelines.**
(DOC)

**S3 File. Inclusivity in global research questionnaire.**
(DOCX)

**S4 File. Additional illustrative quotes for each of the emergent themes (first guide).**
(DOCX)

**S5 File. Anthropometric characteristics of participants.**
(DOC)

**S6 File. Improvement proposal sketches created by the participants.**
(DOC)

**S7 File. Additional illustrative quotes for each of the emergent themes (second guide).**
(DOCX)

## Acknowledgments

We would like to thank to the community leader and the rural farmer community from Yanaca, Apurímac, Perú. We also thank psychologist Enma Maco Condezo (EMC) for her role in the interviews.

## Author contributions

**Conceptualization:** Liliana Cruz-Ausejo.

**Data curation:** Liliana Cruz-Ausejo.

**Formal analysis:** Liliana Cruz-Ausejo, Jerome T Galea.

**Funding acquisition:** Liliana Cruz-Ausejo.

**Investigation:** Liliana Cruz-Ausejo, MD Zahid Hasan, Amit Bhattacharya, Jerome T Galea.

**Methodology:** Liliana Cruz-Ausejo, Jose del Carmen Abad Castillo, MD Zahid Hasan, Jerome T Galea.

**Project administration:** Liliana Cruz-Ausejo, Jose del Carmen Abad Castillo, Claudia Cardenal.

**Resources:** Liliana Cruz-Ausejo, Jose del Carmen Abad Castillo.

**Software:** Liliana Cruz-Ausejo.

**Supervision:** Liliana Cruz-Ausejo, Jose del Carmen Abad Castillo, Claudia Cardenal, MD Zahid Hasan, Amit Bhattacharya, Jerome T Galea.

**Validation:** Liliana Cruz-Ausejo, Amit Bhattacharya, Jerome T Galea.

**Visualization:** Claudia Cardenal, Jerome T Galea.

**Writing – original draft:** Liliana Cruz-Ausejo, Jose del Carmen Abad Castillo, Claudia Cardenal, MD Zahid Hasan, Amit Bhattacharya, Jerome T Galea.

**Writing – review & editing:** Liliana Cruz-Ausejo, Jose del Carmen Abad Castillo, Claudia Cardenal, MD Zahid Hasan, Amit Bhattacharya, Jerome T Galea.

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
