## [Decision Letter · Decision Letter 0]

4 Jan 2026

PONE-D-25-54226Andean agriculture and hand tools: a qualitative approach of exploration of needs, barriers, and opportunities for innovationPLOS One

Dear Dr. Cruz-Ausejo Liliana,

Thank you for submitting your manuscript to PLOS ONE. After careful consideration, we feel that it has merit but does not fully meet PLOS ONE’s publication criteria as it currently stands. Therefore, we invite you to submit a revised version of the manuscript that addresses the points raised during the review process.

We look forward to receiving your revised manuscript.

Kind regards,

In-Ju Kim, Ph.D.

Academic Editor

PLOS One

Journal Requirements:

“This work was funded by the Consejo Nacional de Ciencia, Tecnología e Innovación Tecnológica (CONCYTEC) and the Programa Nacional de Investigación Científica y Estudios Avanzados (PROCIENCIA) within the framework of the E077-2023-01-BM “Becas en Programas de Doctorado en Alianzas Interinstitucionales” contest, grant number (PE501090201-2024-PROCIENCIA-BM) and the E033-2023-01-BM “Alianzas Interinstitucionales para Programas de Doctorado”, grant number N° PE5010843066-2023-PROCIENCIA-BM).

We would like to thank psychologist Enma Maco Condezo (EMC) for her role in the interviews.”

“L.C-A was funded by the Consejo Nacional de Ciencia, Tecnología e Innovación Tecnológica (CONCYTEC) and the Programa Nacional de Investigación Científica y Estudios Avanzados (PROCIENCIA) within the framework of the E077-2023-01-BM “Becas en Programas de Doctorado en Alianzas Interinstitucionales” contest, grant number (PE501090201-2024-PROCIENCIA-BM) and the E033-2023-01-BM “Alianzas Interinstitucionales para Programas de Doctorado”, grant number N° PE5010843066-2023-PROCIENCIA-BM). The sponsor did not play any role in the study design, data collection and analysis, publication decision, or manuscript preparation. “

Additional Editor Comments :

Thank you for submitting your manuscript (PONE-D-25-54226) entitled "Andean agriculture and hand tools: a qualitative approach of exploration of needs, barriers, and opportunities for innovation" to PLOS ONE.

The manuscript has now been evaluated by two expert reviewers. Both reviewers recognise the novelty, relevance, and potential contribution of the study, particularly in documenting farmer perspectives on hand tool use in Andean agriculture, an underrepresented area within occupational health, ergonomics, and innovation studies. However, both reviewers raise substantial methodological, analytical, and interpretive concerns that must be addressed before the manuscript can be considered for publication.

On this basis, a major revision is required to incorporate these changes. These changes will enhance the manuscript's alignment with PLOS ONE's criteria for technical soundness, reproducibility, and broad relevance. Please address the following key points in your revision:

1) Methodological enhancements: Clarify the two interview guides' objectives, differences, and links to research questions in the Methods section. Defend the focus on potato harvesting. Specify if themes emerged from coding or guides.

2) Ergonomic and quantitative integration: Include tool dimensions (weight, length, size) and ergonomic analysis to address health/MSD issues from the introduction. Quantify findings (e.g., theme frequencies) to make results more robust, accounting for gender differences.

3) Results and visualisation: Enhance tool modification sketches (e.g., via CAD). Avoid quote repetition. Clarify ambiguities in language and consolidate related findings (e.g., weight preferences).

4) Discussion and broader implications: Base discussions on quantified results. Expand on shifts to manufactured tools, mechanisation impacts, blacksmith decline, and repair sustainability. Link tradeoffs (productivity vs. comfort) explicitly to data. Rewrite the conclusion to clearly highlight the major findings.

5) Data sharing: Anonymise and share source testimonies per PLOS ONE policy. Explain any ethical constraints.

6) Minor edits: Correct terminology, rephrase for neutrality, avoid overgeneralization, and place figures in-text.

Provide a point-by-point response to these comments and the original reviewer’ details. We look forward to your revised submission.

Reviewers' comments:

Reviewer's Responses to Questions

**Comments to the Author**

1. Is the manuscript technically sound, and do the data support the conclusions?

Reviewer #1: Partly

Reviewer #2: Yes

2. Has the statistical analysis been performed appropriately and rigorously? 

Reviewer #1: N/A

Reviewer #2: N/A

3. Have the authors made all data underlying the findings in their manuscript fully available?

Reviewer #1: Yes

Reviewer #2: No

4. Is the manuscript presented in an intelligible fashion and written in standard English?

Reviewer #1: Yes

Reviewer #2: Yes

5. Review Comments to the Author

Reviewer #1: Abstract: Concise and well-formed

Introduction: well written

Material and methods: nicely explained, but no scientific approach is used to collect data.

Results: Tables and figures are elaborated very well. Each head was clearly explained. The statements made by participants clearly depict the existing situation of respondents by using hand tools. It is a detailed qualitative approach for analyzing hand tools.

However;

In the introduction, health hazards and MSDs caused by the use of poorly designed hand tools were discussed. But in the study, the ergonomic points of the tools' analysis were missing

Since no statistical tool has been employed to infer the findings, the results do not appear to be statistically sound.

· Health risks and MSDs brought on by using a hand tool with inadequate design were covered in the introduction. However, in the study, ergonomic tool analysis is absent.

Second, the results are not inferred using any statistical tools. As a result, the study's overall findings are lacking. The result should focus on some clear and quantifiable information, like which parts of the hand tool were deemed to have poor ergonomic design or require revision.

Further, both men and women were included in the study as samples; however, they differ greatly in their physical characteristics and how they use hand tools.

· In the results, dimensions of hand tools are missing, which are necessary for assessing the tool, such as weight, length, and size. It is not ergonomically sound to analyze tools just based on statements

· Farmer-provided or farmer-made sketches for modifying the current tool design are unclear. All recommendations for creating or making changes to the tool design can be included in a sketch or drawing (in CAD). It will be a meaningful finding and feedback for further investigation.

Discussion: Comprehensively written by detailing each component.

But the discussion is only made based on statements of farmers, not on the crux of the results. A statement made by one farmer does not reflect the opinion/point of view of other farmers. So there should be some quantification of data/results to interpret the findings in a meaningful manner.

Conclusion: Not explaining the exact picture of the result. It should be rewritten to highlight the major findings of the study.

Supporting information: The detailing of each component is added and clearly exhibits the research.

Reviewer #2: The paper is a quite focused study of farmer attitudes and perceptions regarding their hand tools in a Peruvian Andean community. It is a bit reduced in terms of overall scope and sample size, the latter of which the authors acknowledge. It would have been nice to do the study across a number of communities with a smaller set of farmers in each community, to increase the generalizable nature of the results. However as this is a relatively novel aspect of a more general occupational health/literature, there is something to be learned here. My two main critiques of the paper and reason for major revisions are first, that the methods and especially the survey guides need to be more clearly explained in terms of their objectives, link to the research questions, differentiation among the two, and description of the areas covered or the technique. Sometimes the objective of a guide is only found in the results, very late for building up the understanding of the methodology. Without this, the thematic areas of the results seem rather haphazard.

Second, within the limitations of a small and potentially overly ‘unique’ (ie one village) sample size which is already acknowledged, I would urge the authors to expand a bit on the wider meaning and implications of the results. What about the observation that manufactured tools are increasingly prevalent and older, bespoke tools are increasingly rare? Are there implications from increasing mechanization and demographic change?

In general, there is a need for a bit more detail in certain areas of the paper where the logical flow or background to information is missing, as is noted below in the detailed line by line comments.

Also, for ease of reviewing if I review this manuscript again, I would ask that the figures be placed within the text, it makes it much easier and actually increases the speed and quality of the review.

One last point, I have indicated that I don’t feel this paper meets the data sharing standards of PLOS One. That is because individual access with consent of the authors to the data is not really essential in my view. It should be possible to anonymize the source testimony material and erase any personal mentions such that the database would remain as a textual resource showing in a more detailed way all the attitudes towards tools and their context. If the ethics board is denying this option then the authors may claim so, and perhaps the main editor will disagree, but I would urge you just to prepare a text resource in this way, with no personal or geographic information, and share, even in the online materials. That is very broadly in line with ethics standards, especially as these are opinions about management and tools rather than personal, health, or family details.

Detailed comments with line numbers:

47 ecological variability not variety

55 Looking at your references here, I think that you should approach the statement “many of which lack ergonomic principles” with more skepticism. I am wondering whether you are conflating lack of ergonomic principles with lack of mechanization which could lead to less overall effort and fatigue? At least in some of the Andean contexts where I have worked, and in fact as you document in this paper, farmers indicate strong preference for certain of their tools and even will approach blacksmiths with particular design elements that make tools more suited to their work setting. However this may be different in Peruvian Andean contexts if all tools are increasingly manufactured outside (e.g. Brazil, China) and imported to the region – but perhaps separating effort, fatigue, and efficiency is important. you may be able to use the literature on hand tool use from beyond the region that you already cite, that will speak to this point?

62 This is a very generic reference to a textbook (Harrington 2000, Occupational Health) for the notion that tool redesign could help farmers --- could you provide a more specific reference focused on farm environments, or examples of tool design that id in fact help farmers, if you mean to link this to literature?

67 I would say “ innovating further on technological solutions “ rather than “Developing”--- the hand tools analysed are in fact already a technological solution, sometimes developed by farmers, and you are making the point here that their context and knowledge should be respected!

71-72 more details on this design model? It’s not commonly known, e.g. what do the “two diamonds” refer to? An example of more detail being needed to fully understand your objectives – a few sentences

75 are these the two objectives of the two instruments below in methods? Do a better job identifying how the methods address these two main questions, then. Also, I am thinking that the first question is more general around needs and context, and the second is more specific around their specific reactions to a set of tools in harvesting. If that is the case, use that terminology (general/context vs. specific etc.) to aid the reader’s understanding.

77 I think you need defend the choice of one particular operation of potato harvesting. Why choose that compared to say, carrot peeling or compost pile turning – tools are used for all of these? I am assuming the choice of potato harvesting was not just opportunistic? It's a pretty central focus of agriculture in Peru etc.

83 – you mention two guides but don’t say how they differed, what was the purpose of each, why two? - if that is the meaning of “and” later in the sentence, enumerate these as the two purposes. More detail and clarity needed.

142 “Data are presented according to the interview guide used” but I don’t understand the difference between the two interview guides, as the methods are incomplete, and here just below you say the first interview guide described perceptions, but again it’s not clear and doesn’t distinguish this from the second guide.

151 Task category – is this a structure from the interview guide, or an emergent theme from coding?—this all needs to be presented more clearly. IPerhaps linking to the figures with the different guide numbers to a greater degree within the text will help?

184 adapt as in choosing different alternatives, or adapt as in modifying their tools? Two different things. Again, the language here is not very specific and seems to lack the complexity that is promised in the introduction.

194 here I would say just children, without the definite article The – we are not talking about particular children.

196 Is this really true that there are no individual plots? Is this a communal sectoral fallow system with use rights granted by a communal entity like a consejo? Because in many areas of Peru labor exchanges may be practiced (some labor is thought of in a common way) but in fact the plots are held privately by families. It would be good to clarify this so that you are not propagating a vague or romantic image of communities holding land in common, if it is not the case, to complement what the farmer said. Maybe clarify this using a comment on this quotation.

221 it is always 7 days, or just as reported by the participants in this study? – or that this is somehow important to the particular focus on the design of hand tools? I find it hard to believe that some other factors like weather or yield levels might not make it last more or less time? In general this seems to reveal a weakness of doing a study in one community in one year, it needs to be carefully presented to not overgeneralize, and it would have been nice to have participants from a range of communities instead of just one, seems like pseudoreplication of sorts.

231 What should resemble the pick? Use parentheses to identify the referent.

247 this actually parallels what I observed in Bolivia in the 2000s, both allachu (called wiluku, there) and picks were used. Given the emphasis on physical lightness of the tools, did no one comment on the fact that a pick is more versatile with two sides but in fact obligates one to carry the weight of a function that is not always used, versus having more or less pointed allachus that are lighter and perhaps more adapted to to the main task of potato harvesting? Also, is the soil in this community rocky, so that extracting or moving rocks is such an important task?

248 is the prevalence of blacksmiths decreasing or staying the same? Did you ask about this? It seems like if blacksmithing skills are lost this would be a natural explanation for the transition to different pick tools that come from outside, and this could be an implication of the study, a need to support more local tradespeople or accomplish that function within local agencies or development organizations?

275 “3.0+/- 0.2 kg” is the +/- a standard error, standard deviation range mentioned by interviews, what? Please note quickly.

281 I think that what p5 is describing here is a sort of local optimum of weight, neither too light or too heavy , as you also say in lines 343 – maybe these results on intermediate weight and relation to age and gender could be consolidated.

317 this is the same quote as in the previous section that presumably comes from the first interview guide and here the second? It seems a bit repetitive as it makes the same point.

332 only now you describe what the second interview guide is about! All this should be described clearly in the methods section.

430 Please change to Jain et al.’s (2018) proposal for improving…. To clarify that this is a reference to an existing paper. Also Jain is not a single author, etc.

431 rephrase as “While adding environment as a new theme within their narratives” or similar. The colon is awkward here since only two words follow it. At a minimum you can say “the environmental theme” (and also refer to the figure 5 again with these themes)

446 by multiple heads please clarify whether you mean a tool with two heads (or more?) or just multiple tools with different sized heads to adapt in the field to the soil conditions.

453 I don’t know that you need a reference to explain the connection between humidity, rain and rotting of wood. Better to stay focused on the farmer viewpoints, rotting and weakening is a well known process.

462 Can you specifically point to where in your research or in which aspects farmers identified the tradeoff between productivity and comfort? I am not remembering a farmer talking about this and since it is a bit contentious, would like to have the connection to the testimonies clearly described. Would not an uncomfortable tool fatigue a farmer more and thus lead to a shortened and thus less productive work day? So it needs to be clearly explained.

478 instead of substitutes would say “acts in place of” or “substitutes for” , I slightly prefer the former because it is somewhat more neutral, not privileging formal or written learning pathways.

481 although you seek to correct a bias in favor of formal versus informal intergenerational training in the next sentences, you could strengthen that perspective in this line by saying “ but rather from the absence of training that would accompany an outside tool in a more formalized technology context” ? because I think the formal training route only applies to tools or methods that are brought from outside? (Whether or not they are effective; sometimes outside tools and formal training can be effective, but as you point out local pragmatic learning enhances a sense of ownership of the tool technology)

492 no capital C on characteristics.

502 I’m glad you recognize the sample within one community as a limitation, I agree – better to document this limitation for readers than ignore it.

512-515 You mention the way that a local repair function is an enabler but I think maybe this should be included in the last few lines that are the main prescriptive outcomes of the paper – that is, whether tools are made locally or come from outside, they will be much more valuable in communities like this one if they are repairable and modifiable. As I noted regarding the results section, it might also flag the availability of skilled blacksmiths in these communities or near to them, as a key factor in the sustainability of hand tool use that should be explored in addition to the other factors mentioned.

6. PLOS authors have the option to publish the peer review history of their article (what does this mean?). If published, this will include your full peer review and any attached files.

Reviewer #1: No

Reviewer #2: No

---

## [Author Response · Author response to Decision Letter 1]

12 Feb 2026

LETTER OF RESPONSE

Dear editor and reviewers

We would like to thank you for the suggestions and comments made on this manuscript. We are sure of the improvement of this based on your recommendations.

Below you will find the answer to each point.

Kind regards.

Response: The manuscript has been revised to adjust PLOS ONE´s style requirements.

Response: The inclusivity in global research is added as a Supplemental file-S3

“This work was funded by the Consejo Nacional de Ciencia, Tecnología e Innovación Tecnológica (CONCYTEC) and the Programa Nacional de Investigación Científica y Estudios Avanzados (PROCIENCIA) within the framework of the E077-2023-01-BM “Becas en Programas de Doctorado en Alianzas Interinstitucionales” contest, grant number (PE501090201-2024-PROCIENCIA-BM) and the E033-2023-01-BM “Alianzas Interinstitucionales para Programas de Doctorado”, grant number N° PE5010843066-2023-PROCIENCIA-BM).

Please remove any funding-related text from the manuscript and let us know how you would like to update your Funding Statement.

Response: Thank you for the clarification. The funding statement states as follows:

This work was funded by the Consejo Nacional de Ciencia, Tecnología e Innovación Tecnológica (CONCYTEC) and the Programa Nacional de Investigación Científica y Estudios Avanzados (PROCIENCIA), within the framework of the E077-2023-01-BM “Becas en Programas de Doctorado en Alianzas Interinstitucionales” contest, grant number N° PE501090201-2024-PROCIENCIA-BM and the E033-2023-01-BM “Alianzas Interinstitucionales para Programas de Doctorado”, grant number N° PE5010843066-2023-PROCIENCIA-BM. The sponsor did not play any role in the study design, data collection and analysis, publication decision, or manuscript preparation”

4. We note that you have indicated that there are restrictions on data sharing for this study. PLOS only allows data to be available upon request if there are legal or ethical restrictions on sharing data publicly. For more information on unacceptable data access restrictions, please see http://journals.plos.org/plosone/s/data-availability#loc-unacceptable-data-access-restrictions.

Response: The de-anonymized dataset has been uploaded to the OSF repository and is available at: https://osf.io/fvr9t

Reviewer’s comments

Reviewer #1:

1. Abstract: Concise and well-formed

Introduction: well written

Material and methods: nicely explained, but no scientific approach is used to collect data.

Results: Tables and figures are elaborated very well. Each head was clearly explained. The statements made by participants clearly depict the existing situation of respondents by using hand tools. It is a detailed qualitative approach for analyzing hand tools. However, In the introduction, health hazards and MSDs caused by the use of poorly designed hand tools were discussed. But in the study, the ergonomic points of the tools' analysis were missing. Since no statistical tool has been employed to infer the findings, the results do not appear to be statistically sound. Health risks and MSDs brought on by using a hand tool with inadequate design were covered in the introduction. However, in the study, ergonomic tool analysis is absent Second, the results are not inferred using any statistical tools. As a result, the study's overall findings are lacking. The result should focus on some clear and quantifiable information, like which parts of the hand tool were deemed to have poor ergonomic design or require revision. In the results, dimensions of hand tools are missing, which are necessary for assessing the tool, such as weight, length, and size. It is not ergonomically sound to analyze tools just based on statements

Response: Thank you for the comments. We strengthened the methodology and results section of the manuscript by complementing the qualitative findings with descriptive quantitative synthesis. In the results section, pain-related codes were quantified by body region at the participant level and are now reported as the number and percentage of participants who mentioned pain in each region at least once (presence/absence per interview), using a binarized code–document table in ATLAS.ti v25. These results are now presented in Table 1 (n/N, %), including stratification by sex. In addition, we incorporated an ergonomic description of the hand tools assessed, reporting their physical/ergonomic characteristics in Table 2. And a summary of tool components/parts reported by farmers as problematic. Quotations were retained only as illustrative examples; the interpretation and discussion are now grounded on the quantified results and tool measurements. It is worth mentioning that a more comprehensive ergonomic assessment (anthropometry, observational evaluation such as REBA, biomechanical analysis, and sEMG) is addressed in a separate manuscript; however, this paper remains focused on farmers’ experiences while incorporating the above descriptive quantitative summaries and tool characteristics to support and contextualize the findings.

2. Further, both men and women were included in the study as samples; however, they differ greatly in their physical characteristics and how they use hand tools.

Response: We agree. Men and women can differ in physical characteristics (anthropometry, height, etc.) and in how they use hand tools, which may affect discomfort and tool suitability. We updated the manuscript to present the MSDs outcomes (pain and discomfort) stratified by sex and to interpret these differences in relation to the measured physical/ergonomic characteristics of the tools. Also, we added an anthropometric measurements table in supplementary file 5, to complement and better understand physical characteristics differences among participants. Lines 285-294.

3. Information added: Farmer-provided or farmer-made sketches for modifying the current tool design are unclear. All recommendations for creating or making changes to the tool design can be included in a sketch or drawing (in CAD). It will be a meaningful finding and feedback for further investigation.

Response: We appreciate the suggestion. The sketches produced by the farmers were included as originally made in the Supplementary Material because our aim is to faithfully preserve the feedback generated during the participatory process. These drawings serve as qualitative inputs to identify opportunities for improvement (e.g., adding a cutting edge to the blade/tip, adjustments to weight, length etc.), rather than as proposals with defined dimensions and geometry. For this reason, we did not convert them into CAD in this manuscript. Producing CAD models would require assuming measurements and technical details that are neither contained nor validated by the original sketches and could therefore introduce an artificial level of precision. The translation of these suggestions is undertaken later during the detailed design stage.

4. Discussion: Comprehensively written by detailing each component. But the discussion is only based on statements of farmers, not on the crux of the results. A statement made by one farmer does not reflect the opinion/point of view of other farmers. So, there should be some quantification of data/results to interpret the findings in a meaningful manner.

Response: We agree with this observation. To ensure the discussion is grounded in the core results and not only on individual farmers’ statements, but we also incorporated a quantitative summary of the findings. Specifically, we added results on bodily discomfort and pain reported by farmers stratified by sex, and we present these outcomes in relation to the physical and ergonomic characteristics of the hand tools used. This allows the discussion to reflect patterns across the sample and supports a more meaningful interpretation. Lines 539-570

5. Conclusion: Not explaining the exact picture of the result. It should be rewritten to highlight the major findings of the study.

Response: Thank you for the suggestion. We rewrote the conclusion section to include relevant findings. Lines 661-677

6. Supporting information: The detailing of each component is added and clearly exhibits the research.

Response: Thank you

Reviewer #2.

1. The paper is a quite focused study of farmer attitudes and perceptions regarding their hand tools in a Peruvian Andean community. It is a bit reduced in terms of overall scope and sample size, the latter of which the authors acknowledge. It would have been nice to do the study across a number of communities with a smaller set of farmers in each community, to increase the generalizable nature of the results. However, as this is a relatively novel aspect of a more general occupational health/literature, there is something to be learned here. My two main critiques of the paper and reason for major revisions are first, that the methods and especially the survey guides need to be more clearly explained in terms of their objectives, link to the research questions, differentiation among the two, and description of the areas covered or the technique. Sometimes the objective of a guide is only found in the results, very late for building up the understanding of the methodology. Without this, the thematic areas of the results seem rather haphazard.

Response: Thank you for these comments. We agree that conducting the study across multiple communities would have strengthened generalizability; however, this was not feasible due to limited financial and human resources, and we now state this more explicitly as a study limitation. Regarding the methodology, we revised the Methods to clarify the objectives and scope of each guide, their alignment with the two research questions, and the differences between guide 1 and 2. We strengthen the Data analysis section to make clear how the results were organized: coding was guided by a priori analytic domains (human, product, task, qualitative), with codes developed inductively from the data; an additional environmental domain emerged during coding and was incorporated as a fifth domain. Finally, we state that results are reported according to these five analytic domains. Lines 109-129; 168-180

2. Second, within the limitations of a small and potentially overly ‘unique’ (i.e. one village) sample size which is already acknowledged, I would urge the authors to expand a bit on the wider meaning and implications of the results. What about the observation that manufactured tools are increasingly prevalent and older, bespoke tools are increasingly rare? Are there implications from increasing mechanization and demographic change? In general, there is a need for a bit more detail in certain areas of the paper where the logical flow or background to information is missing, as is noted below in the detailed line by line comments.

Response: Thank you for the comment. We have expanded the Discussion section to improve the logical flow and to situate our results within wider demographic and technological changes occurring in Andean agricultural context. Drawing on field observations, we added a paragraph describing how rural out-migration of younger populations, the aging agricultural workforce, and the availability of imported tools and replacement parts shape farmers’ decisions. Lines 626-638

3. Also, for ease of reviewing if I review this manuscript again, I would ask that the figures be placed within the text, it makes it much easier and actually increases the speed and quality of the review.

Response: Dear reviewer Initially, we included the images within the manuscript as requested; however, according to the PLOS editorial policy, images must be submitted as separate files, which is why we could not finally include them within the manuscript.

4. One last point, I have indicated that I don’t feel this paper meets the data sharing standards of PLOS One. That is because individual access with consent of the authors to the data is not really essential in my view. It should be possible to anonymize the source testimony material and erase any personal mentions such that the database would remain as a textual resource showing in a more detailed way all the attitudes towards tools and their context. If the ethics board is denying this option then the authors may claim so, and perhaps the main editor will disagree, but I would urge you just to prepare a text resource in this way, with no personal or geographic information, and share, even in the online materials. That is very broadly in line with ethics standards, especially as these are opinions about management and tools rather than personal, health, or family details.

Response: Thank you for this recommendation. In line with Plos One data sharing standard, we have prepared a de-identified textual dataset derived from the source testimony material. All direct identifiers and potentially identifying details (e.g., names, personal references) were removed to minimize re-identification. Data is available at: https://osf.io/fvr9t

Detailed comments with line numbers:

47 ecological variability not variety

Response: text corrected. Line 48

55 Looking at your references here, I think that you should approach the statement “many of which lack ergonomic principles” with more skepticism. I am wondering whether you are conflating lack of ergonomic principles with lack of mechanization which could lead to less overall effort and fatigue? At least in some of the Andean contexts where I have worked, and in fact as you document in this paper, farmers indicate strong preference for certain of their tools and even will approach blacksmiths with particular design elements that make tools more suited to their work setting. However, this may be different in Peruvian Andean contexts if all tools are increasingly manufactured outside (e.g. Brazil, China) and imported to the region – but perhaps separating effort, fatigue, and efficiency is important. you may be able to use the literature on hand tool use from beyond the region that you already cite, that will speak to this point?

Response: Thank you for the observation. We revised the text to clarify that Ergonomic adequacy should not be equated with the absence of mechanization, but rather the design of work systems, including tools. We emphasize that manual tools may generate musculoskeletal risk when tool design and task demands are not aligned with human capabilities, even when such tools are locally preferred. In addition, we incorporated evidence from hand tool use in non-agricultural settings to support this distinction. The revisions are included in lines 56-63

62 This is a very generic reference to a textbook (Harrington 2000, Occupational Health) for the notion that tool redesign could help farmers --- could you provide a more specific reference focused on farm environments, or examples of tool design that id in fact help farmers, if you mean to link this to literature?

Response: We thank the reviewer for this comment. The

---

## [Decision Letter · Decision Letter 1]

19 Mar 2026

PONE-D-25-54226R1Andean agriculture and hand tools: a qualitative approach of exploration of needs, barriers, and opportunities for innovationPLOS One

Dear Dr. Liliana Cruz-Ausejo,

Thank you for submitting your manuscript to PLOS ONE. After careful consideration, we feel that it has merit but does not fully meet PLOS ONE’s publication criteria as it currently stands. Therefore, we invite you to submit a revised version of the manuscript that addresses the points raised during the review process.

We look forward to receiving your revised manuscript.

Kind regards,

Prof. In-Ju Kim, Ph.D.

Academic Editor

PLOS One

Journal Requirements:

Additional Editor Comments:

Thank you for the revised manuscript "Andean agriculture and hand tools: a qualitative approach of exploration of needs, barriers, and opportunities for innovation" (PONE-D-25-54226R1).

The manuscript has now been evaluated by two expert reviewers. Both reviewers are satisfied with your responses to prior comments.

The manuscript is now technically sound, the data support the conclusions, the underlying data are available, and the writing is clear.

They endorse publication after minor editorial polishing.

Below, the detailed feedback is provided to substantiate the above decision.

Abstract & Introduction

• Line 35: Add phrase to mark results (e.g., “We found that hand tools used in…”).

• Line 44: Clarify/spell out “MeSH terms.”

• Line 49: Use “diversified smallholder farming” instead of “subsistence systems.”

• Ensure consistent tenses (present for general/intro; past for methods/results).

• Fix typos/transitions: lines 69 (“tools”), 72 (add period), 74–75 (“annual” not “temporary”), 76 (insert word), 78 (“important context”), 88 (add transition to “double diamond”), 93 (“define”).

Methods & Results

• Lines 97, 113, 120, 123, 129, 171, 188, 190, 197, 218: Simplify phrasing, use past tense, parentheses for supplemental refs, rephrase Jain citation, change “de-anonymized” to “anonymized,” add figure spaces.

• Lines 257, 260, 273, 288, 326, 334, 353, 368: Add comma (“collective, is…”), use “private/owned,” remind “allachu (hoe),” simplify fractions, clarify woods, confirm singular/plural, lowercase “barriers and enablers,” consistent “grip diameter.”

• Tables 2 & 3: Remove redundant circumference if equivalent; shorten title; simplify columns (absolute/percentage + denominator in heading).

• Lines 415–416, 446, 447, 474, 489, 491, 515, 538, 548, 559, 579: Clarify rust/soil wear (optional), flag second-question transition, delete redundant “however,” maintain past tense, rephrase unclear sentences (e.g., familiarity, inherent experiences), move site factors (rocky soils) to environment if better fit, add concluding sentence on heavy tools (fatigue vs. soil benefits), fix “total tool length” phrasing.

Discussion & Conclusion

• Lines 562, 579, 605, 613, 622, 633, 655: (Optional) Consider Andean body proportions for tool length; note soil type in wear; “emerged” not “emerges”; nuance to “include” technical criteria (balance with local knowledge); strengthen participatory design transition; add comma; briefly defend sample focus/limitation.

Figures

• Correct spelling/grammar: Fig. 1 (“terrain conditions,” “means of transport,” “discomfort and bodily pain”); Fig. 5 (“deficient”); capitalise first words in lists.

Please address these points in a point-by-point response and submit the final revision.

Reviewers' comments:

Reviewer's Responses to Questions

**Comments to the Author**

1. If the authors have adequately addressed your comments raised in a previous round of review and you feel that this manuscript is now acceptable for publication, you may indicate that here to bypass the “Comments to the Author” section, enter your conflict of interest statement in the “Confidential to Editor” section, and submit your "Accept" recommendation.

Reviewer #1: All comments have been addressed

Reviewer #2: (No Response)

2. Is the manuscript technically sound, and do the data support the conclusions?

Reviewer #1: Yes

Reviewer #2: Yes

3. Has the statistical analysis been performed appropriately and rigorously? 

Reviewer #1: Yes

Reviewer #2: N/A

4. Have the authors made all data underlying the findings in their manuscript fully available?

Reviewer #1: Yes

Reviewer #2: Yes

5. Is the manuscript presented in an intelligible fashion and written in standard English?

Reviewer #1: Yes

Reviewer #2: Yes

6. Review Comments to the Author

Reviewer #1: Abstract: Well-structured and provides a clear overview of the objectives, methodology, major findings, and significance of the study. It effectively summarizes the core aspects of the research and reflects the essence of the manuscript.

Introduction: clearly establishes the importance and relevance of the topic. The background information is well articulated and supported by appropriate and relevant literature. The cited studies help in establishing the research gap and justify the need for the present investigation.

Materials and methods: This section is now systematically organized and clearly presented. The headings and subheadings effectively describe the procedures and methodological approaches adopted for data collection and analysis. The section is sufficiently detailed to allow readers to understand the research design and the processes followed in conducting the study.

Results: The results section reflects the considerable effort made by the researcher to present the findings comprehensively. The major recommendations provided in the earlier review appear to have been incorporated. The findings are well organized and presented in a technically sound manner.

Discussion: The researcher has made a commendable effort to interpret the results and explain the underlying reasons for the observed findings. The discussion appropriately relates the results to the perceptions of respondents regarding agricultural work and the use and application of hand tools in their daily work practices.

Conclusion: Concise and clearly highlights the major findings derived from the study.

Overall, the researcher has made significant efforts to improve the manuscript. The revisions have strengthened the quality and clarity of the paper. In its present form, the manuscript represents a meaningful contribution to the field and will be useful for future researchers working in this domain.

Recommendation

Based on the improvements made and the overall quality of the manuscript, the paper is recommended for publication.

Reviewer #2: I am pleased to say that the authors have addressed the questions in my review and have substantially improved the manuscript. I am only making some detailed suggestions for editing that should be considered to remove errors and make the manuscript more readable. These should take very little time and I then endorse the manuscript for publication.

Detailed comments

35 would suggest inserting here “We found that hand tools used in…” to delineate this section of the abstract as results

44 what are MeSH terms? Maybe spell out or just write them?

49 It may not be germane to the research topic, but especially if you are setting the bar at 97% of farms, I don’t think these are exclusively subsistence systems. As you clarify below, some of their crops are for sale presumably if 66% is for home consumption? It’s not crucial for your study but I would find a different term, perhaps diversified smallholder farming (because income streams are increasingly diverse, not just farming, mining etc.). It doesn’t undermine any of your design or findings, not a dealbreaker so to speak, but is just a chance not to perpetuate myths about Andean farmers.

52 allocate, the previous sentences are in present tense as is appropriate and here past tense is appearing. Please check tenses throughout; intro generally present, methods, results narrated as past; and discussion probably a mix of present and past…I think I found most of the instances of mis-tensed verbs but a check would be good.

69 tools

72 period at end of sentence

74,75, etc. temporary, I would say annual. Temporary does not have that implication in English; annual vs. perennial crops is the usual distinction.

76 I think you are missing a word, national importance or prevalence (to not be redundant with importance below)?

78 I would say important context rather than particular

88 More of a transition is needed, e.g. “To address this need for co-creation, the “double diamond” design model… otherwise too sudden.

93 define not defines

97 I am not sure that the text after the colon is needed, … two questions: 1….2… is sufficiently clear. Was “a contextual question (needs…) “ extra text left after editing? Because it is quite confusing – reword or omit.

113 (same for line 123) Less awkward is simply “Guide 1 addressed research question one…” (past tense needed)

120 put (Q18-19; see supplemental file one) in parentheses at end rather than as a sentence, or “For more details, see supplemental file 1” . more readable.

129 Similar to above, parentheses can be combined as (Q6; supplemental file 1)

171 Jain et al. works – I would rewrite this slightly, probably “works by Jain et al. (e.g. Jain et al., 2018)” – if I see correctly in your discussion the other Jain reference, with at least one key reference for the reader.

188 slight rephrasing ...emerged during analysis (in this case, after the 11th interview).

190 is it not ‘anonymized data?’ de-anonymized would be with the names put back in, not what is wanted!

197 Would just shorten to “ ..is included in Supplemental file 3.”

218 I am not sure it is in the pdf but please check spaces in e.g. Fig. 2, Fig. 3 – should have a space between fig. and number.

257 perhaps better written with a comma, unless I am understanding the farmer wrong: “…is collective, is a communal workday” this reflects better the speech pattern, for me.

260 If they are regulated through titles I think the word in English would be ‘private’ rather than inherent; or “owned” ? Inherent does not have that meaning.

273 could remind the reader of the meaning of allachu here with allachu (hoe) and maybe even (hoe; Fig. 4)

288 I would write simply “…three of 13 participants” in the text (no “each”, or I am somehow not understanding what you mean by each); the table fractions are well understood.

326 I think tasta and chachas are woods? Maybe say (made from tasta or chachas wood) to complete the thought.

334 did the farmer use the singular? They did not say ‘allachus’ ? just checking.

353 (barriers and enablers) no capital B

368 Which diameter? That of the handle? And this was measured where the hand engages it? Below you use the term grip diameter which is very clear, probably say that?

Table 2 – is circumference of grip redundant with diameter of grip (handle), ie. Just multiplied by 3.14156…? if so probably only one is needed, to simplify the table, I would just cite the diameter.

387 you may want to shorten this title somewhat, it usually should not run to two lines in the journal, eg. After the colon: “..weight, size, design and other material characteristics” ?

Table 3 this is a nice addition, gives some very specific design considerations. I wonder whether it is necessary to give the /13 or /7 e.g. in the columns, probably just giving the absolute and percentage is enough, since you give the denominator in the heading. And, in the heading you could just say “ out of 13” e.g.

415 (side topic) I would be interested in knowing if, assuming tools are put away in a relatively dry place, whether it is rust or the physical wear in the soil/rocks (which is definitely apparent in older tools I have seen) that is most important (e.g. for you, line 458). A worked tool quickly regains its shine in the soil eg.

416 Are we switching to the second research question and interview guide now? Would be good to flag that! You say that in line 437, but does it actually begin here and deserves noting?

446 would delete however, redundant with while

447 associated if we are still in results (past tense, not general knowledge)

474 varied, included, -- keep the results in past tense as you are doing throughout moste of these paragraphs.

489 Was this something that became apparent in the fieldwork (Evaluation as the core of satisfaction?) or a general statement coming from your research framework? If the former, maybe “Evaluation emerged in the field interviews as core to … “

491 I don’t quite understand the phrase “though not all are fully accepted” – the assessments? The expectations? Implicit tool criteria? Please reword for clarity.

515 I think you mean “Were inherent”? but in any case the phrase “experiences were inherent to agricultural activity but also became more familiar” is unclear to me, as regards both literal meaning and significance? Can you rephrase? Maybe stressing that the familiarity of the tool is an important component as the quotes seem to suggest?

538 You place only weather impacts on the tool in the environment category; is it possible to place other site factors there that are presently in the ‘task’ category, e.g. terrain conditions like slope, rocky soils that may wear a tool more quickly; arguably the latter is as impactful or more so on the wear of a tool than a rainy climate and rotting handle?

559 “total tool length with the blade placed on the ground should be approximately….” Is the way to write this. Total is repeated and it is unclear.

579 conditions

548 I feel that there may be an unstated conclusion here that you may want to bring out more explicitly, that a) it should be considered that tools are too heavy in design work going forward because they create too much fatigue but b) there may be reasons for a heavy tool in heavy soils, drawn from your participatory design effort. I wonder if a concluding sentence to the paragraph could say something like this.

562 Have you considered the literature regarding how Andean highland conditions may have produced human evolution towards shortened extremities (for heat conservation, or also unfortunately as a response to stress in childhood e.g. https://journals.plos.org/plosone/article?id=10.1371/journal.pone.0051795 )? It may be worth looking at, I don’t know if there has been a generational fading of this body shape pattern but it was definitely the case in my experience that Andean craft knitters had to deliberately lengthen the sleeves of sweaters for Northern tourists in past decades, if they used local arm models, sleeves were too short. This would make elbow height an unreliable guide and might produce tools that are too long and potentially more tiring; i.e. hard to swing; there may be an adaptedness of a shorter tool for shorter arms, in spite of bending tradeoffs. More generally considering whether the principles of tool measures from outside the region may be valid but the exact values might not be?

579 again soil type might be an additional factor accelerating wear of the tool tip?

605 I would say emerged not emerges (because in this sentence you are referring to the second question and therefore particular instance of results rather than generality)

613 Perhaps overstating the point, but I wonder if the conclusion here should be more nuanced in line with the other elements of the discussion such as in the next paragraph; they should perhaps ‘include technical criteria’ because the other principle you are demonstrating in the discussion is the local knowledge and adaptations of tool users? Rather than ‘adhering to’ which suggests that farmers may have a higher risk of getting it wrong. In my experience farmers who have enough experience to repair or modify tools often have a great deal of understanding of the particular problem they are trying to solve and are more often successful (and also line up well with technical criteria)

622 this is a nice point and balances well against the rest of the paragraph. It might be a chance to stress the importance of participatory design (e.g. double diamond?) that ignores neither local knowledge nor the potential improvements from innovation with design criteria? – and perhaps a good transition to the next paragraph which deepens this argument?

633 would insert a comma “..older, locally crafted implements, …”

655 I think you could insert a short phrase here defending the fact that you chose high frequency of use tools within a major and relatively universal Andean cropping activity (potato harvesting) --- eg. “ Although we examined tools that are ubiquitous in the Andes and a corresponding activity that is one of the more common Andean smallholder farm activities, the small sample may limit…”

Figures – some errors in spelling:

Fig. 1: check these: terrain conditions, means of transport not mean, discomfort and bodily pain,

Fig. 5: deficient tool not defficient; check that all bulleted entries on lists in boxes are capitalized on first word

7. PLOS authors have the option to publish the peer review history of their article (what does this mean?). If published, this will include your full peer review and any attached files.

Reviewer #1: No

Reviewer #2: No

---

## [Author Response · Author response to Decision Letter 2]

25 Mar 2026

LETTER OF RESPONSE

Dear editor and reviewers

We would like to thank you for the suggestions and comments made on this manuscript. We are sure of the improvement of this based on your recommendations.

Below you will find the answer to each point.

Kind regards.

Reviewer´s comments

Reviewer #1:

1. Abstract: Well-structured and provides a clear overview of the objectives, methodology, major findings, and significance of the study. It effectively summarizes the core aspects of the research and reflects the essence of the manuscript.

Introduction: clearly establishes the importance and relevance of the topic. The background information is well articulated and supported by appropriate and relevant literature. The cited studies help in establishing the research gap and justify the need for the present investigation.

Materials and methods: This section is now systematically organized and clearly presented. The headings and subheadings effectively describe the procedures and methodological approaches adopted for data collection and analysis. The section is sufficiently detailed to allow readers to understand the research design and the processes followed in conducting the study.

Results: The results section reflect the considerable effort made by the researcher to present the findings comprehensively. The major recommendations provided in the earlier review appear to have been incorporated. The findings are well organized and presented in a technically sound manner.

Discussion: The researcher has made a commendable effort to interpret the results and explain the underlying reasons for the observed findings. The discussion appropriately relates the results to the perceptions of respondents regarding agricultural work and the use and application of hand tools in their daily work practices.

Conclusion: Concise and clearly highlights the major findings derived from the study.

Overall, the researcher has made significant efforts to improve the manuscript. The revisions have strengthened the quality and clarity of the paper. In its present form, the manuscript represents a meaningful contribution to the field and will be useful for future researchers working in this domain.

Recommendation

Based on the improvements made and the overall quality of the manuscript, the paper is recommended for publication.

Response: Thank you for the comments.

Reviewer #2:

I am pleased to say that the authors have addressed the questions in my review and have substantially improved the manuscript. I am only making some detailed suggestions for editing that should be considered to remove errors and make the manuscript more readable. These should take very little time and I then endorse the manuscript for publication.

Detailed comments

1. 35 would suggest inserting here “We found that hand tools used in…” to delineate this section of the abstract as results

Response: Corrected. Line 35-36

2. 44 what are MeSH terms? Maybe spell out or just write them?

Response: Thank you for the comment. MeSH (Medical Subject Headings) terms are standardized keywords used in biomedical databases to index and organize articles. The term has been spelled out. Line 44.

3. 49 It may not be germane to the research topic, but especially if you are setting the bar at 97% of farms, I don’t think these are exclusively subsistence systems. As you clarify below, some of their crops are for sale presumably if 66% is for home consumption? It’s not crucial for your study but I would find a different term, perhaps diversified smallholder farming (because income streams are increasingly diverse, not just farming, mining etc.). It doesn’t undermine any of your design or findings, not a dealbreaker so to speak, but is just a chance not to perpetuate myths about Andean farmers.

Response: Thank you for the observation. We agree that the term “subsistence family farming” may not fully capture the complexity of these farming systems. Accordingly, we have revised the text to use “diversified smallholder family farming,” which better reflects the combination of self-consumption and market production, as well as the diversity of income sources among these households. Lines 49-50, 54.

4. 52 allocate, the previous sentences are in present tense as is appropriate and here past tense is appearing. Please check tenses throughout; intro generally present, methods, results narrated as past; and discussion probably a mix of present and past…I think I found most of the instances of mis-tensed verbs but a check would be good.

Response: Thank you for the comment, we checked the tense time use throughout the manuscript

5. 69 tools

Response: Corrected. Line 66.

6. 72 period at end of sentence

Response: Corrected. Line 72

7. 74,75, etc. temporary, I would say annual. Temporary does not have that implication in English; annual vs. perennial crops is the usual distinction.

Response: Thank you for this clarification. We corrected the term. Line 74

8. 76 I think you are missing a word, national importance or prevalence (to not be redundant with importance below)?

Response: Thank you for the observation. We added the missing word “relevance”. Line 76.

9. 78 I would say important context rather than particular

Response: Thank you for the observation. We corrected the term to clarify the idea. Line 78

10. 88 More of a transition is needed, e.g. “To address this need for co-creation, the “double diamond” design model… otherwise too sudden.

Response: Thank you for this comment, we corrected the phrase to improve transition. Line 88.

11. 93 define not defines

Response: Corrected. Line 93.

12. 97 I am not sure that the text after the colon is needed, … two questions: 1….2… is sufficiently clear. Was “a contextual question (needs…) “ extra text left after editing? Because it is quite confusing – reword or omit.

Response: Thank you for the comment. We have removed the additional text after the colon, as it was unnecessary and caused confusion.

13. 113 (same for line 123) Less awkward is simply “Guide 1 addressed research question one…” (past tense needed)

Response: Corrected. Lines 113, 123

14. 120 put (Q18-19; see supplemental file one) in parentheses at end rather than as a sentence, or “For more details, see supplemental file 1”. more readable.

Response: Thank you for this suggestion. We have revised the text by placing the reference to the supplemental file in parentheses at the end of the sentence to improve readability. Line 120.

15. 129 Similar to above, parentheses can be combined as (Q6; supplemental file 1)

Response: Corrected. Line 129.

16. 171 Jain et al. works – I would rewrite this slightly, probably “works by Jain et al. (e.g. Jain et al., 2018)” – if I see correctly in your discussion the other Jain reference, with at least one key reference for the reader.

Response: Thank you for this suggestion. We have revised the text to “works by Jain et al.” and included a representative reference (Jain et al., 2018). Line 171,172.

17. 188 slight rephrasing ...emerged during analysis (in this case, after the 11th interview).

Response: Thank you for the suggestion. We rephrased the sentence. Line 188,189

18. 190 is it not ‘anonymized data?’ de-anonymized would be with the names put back in, not what is wanted!

Response: Thank you for this observation. We corrected the term to: “anonymized data”. Line 190.

19. 197 Would just shorten to “ ..is included in Supplemental file 3.”

Response: Corrected. Line 197.

20. 218 I am not sure it is in the pdf but please check spaces in e.g. Fig. 2, Fig. 3 – should have a space between fig. and number.

Response: We checked the documented and corrected the spaces.

21. 257 perhaps better written with a comma, unless I am understanding the farmer wrong: “…is collective, is a communal workday” this reflects better the speech pattern, for me.

Response: We added a comma to clarify the speech pattern. Line 256.

22. 260 If they are regulated through titles I think the word in English would be ‘private’ rather than inherent; or “owned” ? Inherent does not have that meaning.

Response: We agree that, so we changed the term to “owned”. Line 259.

23. 273 could remind the reader of the meaning of allachu here with allachu (hoe) and maybe even (hoe; Fig. 4).

Response: Thank you for this suggestion. We have clarified the term by adding “allachu (hoe; Fig. 4)” to remind the reader of its meaning. Line 272.

24. 288 I would write simply “…three of 13 participants” in the text (no “each”, or I am somehow not understanding what you mean by each); the table fractions are well understood.

Response: Corrected. Line 287.

25. 326 I think tasta and chachas are woods? Maybe say (made from tasta or chachas wood) to complete the thought.

Response: That´s correct. We added “wood” to complete the thought. Line 326.

26. 334 did the farmer use the singular? They did not say ‘allachus’ ? just checking.

Response: that’s correct, the farmer refers to allachu in singular.

27. 353 (barriers and enablers) no capital B

Response: Corrected. Line 354.

28. 368 Which diameter? That of the handle? And this was measured where the hand engages it? Below you use the term grip diameter which is very clear, probably say that?

Response: Thank you for this comment. We have clarified the terminology to avoid confusion. “Handle diameter” is used to refer to the tool measurement, whereas “grip diameter” refers to the user’s hand measurement. Accordingly, we specify “the average handle diameter of both tools…” and, in the following lines, “the maximum grip diameter for women was…”. This distinction helps to clearly differentiate between both terms. Line 369.

29. Table 2 – is circumference of grip redundant with diameter of grip (handle), ie. Just multiplied by 3.14156…? if so probably only one is needed, to simplify the table, I would just cite the diameter.

Response: Thank you for this observation. While circumference grip and handle diameter are not identical measures, they are directly related and therefore convey overlapping information. To simplify the table and avoid redundancy, we have removed the circumference grip measure and retained only the handle diameter.

30. 387 you may want to shorten this title somewhat, it usually should not run to two lines in the journal, eg. After the colon: “..weight, size, design and other material characteristics” ?

Response: Thank you for this suggestion. We have shortened the title to improve readability while preserving the key elements. Now it says: “product consideration: weight, size, handle, material characteristics and durability”. Line 388,389.

31. Table 3 this is a nice addition, gives some very specific design considerations. I wonder whether it is necessary to give the /13 or /7 e.g. in the columns, probably just giving the absolute and percentage is enough, since you give the denominator in the heading. And, in the heading you could just say “ out of 13” e.g.

Response: Thank you for this suggestion. We have revised Table 3 by removing the denominators from the cells and indicating the sample size in the column headings.

32. 415 (side topic) I would be interested in knowing if, assuming tools are put away in a relatively dry place, whether it is rust or the physical wear in the soil/rocks (which is definitely apparent in older tools I have seen) that is most important (e.g. for you, line 458). A worked tool quickly regains its shine in the soil eg.

Response: Thank you for this interesting observation. In our field data, participants primarily referred to climatic factors such as humidity and rainfall as contributors to tool deterioration. However, we agree that mechanical wear resulting from repeated contact with soil and rocks is likely to play a significant role, particularly under intensive use. While this aspect did not emerge explicitly in participants’ narratives, we have incorporated a brief reflection in the discussion to acknowledge that tool degradation may result from both environmental exposure and abrasive interactions during use.

33. 416 Are we switching to the second research question and interview guide now? Would be good to flag that! You say that in line 437, but does it actually begin here and deserves noting?

Response: Thank you for this observation. The transition to the second research question is introduced later in the text. To avoid confusion, we have made this transition more explicit by clearly indicating the shift to the second research question and interview guide.

Now it says: “The following section addressed the second research question, aimed at understanding the perceived satisfaction of farmers regarding the use of traditional hand tools. The resulting categories and codes are presented in Fig. 5, and additional representative quotes are presented in Supplemental file 7”. Lines 438-441.

34. 446 would delete however, redundant with while

Response: Corrected. Line 447.

35. 447 associated if we are still in results (past tense, not general knowledge)

Response: Corrected. Line 449.

36. 474 varied, included, -- keep the results in past tense as you are doing throughout most of these paragraphs.

Response: Thank you for this observation. We have revised the verb tense in the Results section to ensure consistency and maintain the use of past tense throughout.

37. 489 Was this something that became apparent in the fieldwork (Evaluation as the core of satisfaction?) or a general statement coming from your research framework? If the former, maybe “Evaluation emerged in the field interviews as core to … “

Response: Thank you for this observation. This aspect emerged from fieldwork. We have revised the text to clarify that evaluation was identified through participants’ accounts as a core aspect of satisfaction. Now it says: “Evaluation emerged in the field interviews as a core aspect of farmers’ satisfaction, encompassing both functional performance and the comfort/displeasure using hand tools…” Line 491,492.

38. 491 I don’t quite understand the phrase “though not all are fully accepted” – the assessments? The expectations? Implicit tool criteria? Please reword for clarity.

Response: Thank you for this observation. We have reworded the sentence to clarify that tool acceptance was not uniform and depended on specific design features, such as the shape and width of the tip, which influenced the ease of use. Now it says: “Positive assessments arose when tools met user’s expectations and working conditions. However, tools were not uniformly accepted, as specific design features appeared to influence the performance and ease of use”. Lines 493-495.

39. 515 I think you mean “Were inherent”? but in any case the phrase “experiences were inherent to agricultural activity but also became more familiar” is unclear to me, as regards both literal meaning and significance? Can you rephrase? Maybe stressing that the familiarity of the tool is an important component as the quotes seem to suggest?

Response: Thank you for this observation. We have rephrased the sentence to improve clarity and emphasize that familiarity with the tool, developed through repeated use. Now it says: “This category encompassed farmers’ narratives regarding the prolonged use of their tools and the relationship they develop with them over time. As they consistently use their tools, farmers develop increasing familiarity with their tools, which shapes their confidence and preference during use”. Line 516-519.

40. 538 You place only weather impacts on the tool in the environment category; is it possible to place other site factors there that are presently in the ‘task’ category, e.g. terrain conditions like slope, rocky soils that may wear a tool more quickly; arguably the latter is as impactful or more so on the wear of a tool than a rainy climate and rotting handle?

Response: Thank you for this insightful observation. Terrain conditions were indeed identified by participants; however, they were primarily described in relation to task demands, such as physical effort and ease of tool use, rather than their impact on tool wear. For this reason, they were categorized under the “task” domain. The classification in this study reflects how participants framed their ex

---

## [Editor Report · Decision Letter 2]

1 Apr 2026

Andean agriculture and hand tools: a qualitative approach of exploration of needs, barriers, and opportunities for innovation

PONE-D-25-54226R2

Dear Dr. Cruz-Ausejo Liliana,

We’re pleased to inform you that your manuscript has been judged scientifically suitable for publication and will be formally accepted for publication once it meets all outstanding technical requirements.

Kind regards,

In-Ju Kim, Ph.D.

Academic Editor

PLOS One

Additional Editor Comments (optional):

Thank you for submitting the revised version of your manuscript entitled "Andean agriculture and hand tools: a qualitative approach of exploration of needs, barriers, and opportunities for innovation" (PONE-D-25-54226R2) to PLOS ONE.

I have now received the reports from both reviewers. Reviewer 1 is fully satisfied with the revisions and recommends publication. Reviewer 2 confirms that the authors have adequately addressed all previous comments, with only minor editorial polishing needed. The authors’ detailed point-by-point response shows that every suggestion has been carefully implemented.

Both reviewers agree that the manuscript is now technically sound, that the data support the conclusions, that all underlying data are fully accessible (de-anonymised transcripts in the OSF repository), and that the writing is clear and in standard English. The study makes a meaningful contribution to the fields of participatory design, ergonomics, and occupational health in Andean smallholder agriculture.

I am pleased to accept your manuscript for publication in PLOS ONE pending any final minor production checks. We appreciate the thorough revisions you made in response to the reviewers’ constructive feedback.

Congratulations on the acceptance of your work.
---

## [Editor Report · Acceptance letter]

PONE-D-25-54226R2

PLOS One

Dear Dr. Cruz-Ausejo,

I'm pleased to inform you that your manuscript has been deemed suitable for publication in PLOS One. Congratulations! Your manuscript is now being handed over to our production team.

Kind regards,

on behalf of

Prof. In-Ju Kim

Academic Editor

PLOS One